# IPM-LSTM: A Learning-Based Interior Point Method for Solving Nonlinear Programs

Xi Gao[1], Jinxin Xiong[2,3], Akang Wang[2,3,*], Qihong Duan[1], Jiang Xue[1,*], and Qingjiang Shi[2,4]

[1]School of Mathematics and Statistics, Xi'an Jiaotong University, Xi'an, China
[2]Shenzhen Research Institute of Big Data, China
[3]School of Data Science, The Chinese University of Hong Kong, Shenzhen, China
[4]School of Software Engineering, Tongji University, Shanghai, China

## Abstract

Solving constrained nonlinear programs (NLPs) is of great importance in various domains such as power systems, robotics, and wireless communication networks. One widely used approach for addressing NLPs is the interior point method (IPM). The most computationally expensive procedure in IPMs is to solve systems of linear equations via matrix factorization. Recently, machine learning techniques have been adopted to expedite classic optimization algorithms. In this work, we propose using Long Short-Term Memory (LSTM) neural networks to approximate the solution of linear systems and integrate this approximating step into an IPM. The resulting approximate NLP solution is then utilized to warm-start an interior point solver. Experiments on various types of NLPs, including Quadratic Programs and Quadratically Constrained Quadratic Programs, show that our approach can significantly accelerate NLP solving, reducing iterations by up to $60\%$ and solution time by up to $70\%$ compared to the default solver.

## 1 Introduction

Constrained *Nonlinear Programs* (NLPs) represent a category of mathematical optimization problems in which the objective function, constraints, or both, exhibit nonlinearity. Popular NLP variants encompass *Quadratic Programs* (QPs), *Quadratically Constrained Quadratic Programs* (QCQPs), semi-definite programs, among others. These programs are commonly classified as convex or non-convex, contingent upon the characteristics of their objective function and constraints. The versatility of NLPs allows for their application across a wide array of domains, including power systems (Conejo and Baringo, 2018), robotics (Schaal and Atkeson, 2010), and wireless communication networks (Chiang, 2009).

The primal-dual *Interior Point Method* (IPM) stands as a preeminent algorithm for addressing NLPs (Nesterov and Nemirovskii, 1994; Nocedal and Wright, 1999). It initiates with an infeasible solution positioned sufficiently far from the boundary. Subsequently, at each iteration, the method refines the solution by solving a system of linear equations, thereby directing it towards the optimal solution. Throughout this iterative process, the algorithm progresses towards feasibility and optimality while keeping the iterate well-centered, ultimately converging to the optimal solution. However, a notable computational bottleneck arises during the process of solving linear systems, necessitating matrix decomposition with a runtime complexity of $\mathcal{O}(n^3)$.

Recently, the *Learning to Optimize* (L2O) (Bengio et al., 2021; Chen et al., 2024; Gasse et al., 2022) paradigm has emerged as a promising methodology for tackling various optimization problems,

---

*Corresponding authors: Akang Wang <wangakang@sribd.cn>, Jiang Xue <x.jiang@xjtu.edu.cn>

38th Conference on Neural Information Processing Systems (NeurIPS 2024).

Figure 1: An illustration of the IPM-LSTM approach.

spanning unconstrained optimization (Chen et al., 2022a), linear optimization (Chen et al., 2022b; Li et al., 2024), and combinatorial optimization (Baker, 2019; Gasse et al., 2022; Han et al., 2023). Its ability to encapsulate common optimization patterns renders it particularly appealing. Noteworthy is the application of learning techniques to augment traditional algorithms such as the gradient descent method (Andrychowicz et al., 2016), simplex method (Liu et al., 2024), and IPM (Qian et al., 2024).

We observe that existing works on learning-based IPMs primarily concentrate on solving LPs (Qian et al., 2024). Motivated by the robustness and efficiency of IPMs for general NLPs, we pose the following question:

*Can we leverage L2O techniques to expedite IPMs for NLPs?*

In this study, we propose the integration of *Long Short-Term Memory* (LSTM) neural networks to address the crucial task of solving systems of linear equations within IPMs, introducing a novel approach named IPM-LSTM. An illustration of the IPM-LSTM approach is depicted in Figure 1. Specifically, we substitute the conventional method of solving linear systems with an unconstrained optimization problem, leveraging LSTM networks to identify near-optimal solutions for the latter. We integrate a fixed number of IPM iterations into the LSTM loss function and train these networks within the self-supervised learning framework. The substitution is embedded within a classic IPM to generate search directions. Ideally, the primal-dual solution provided by IPM-LSTM should be well-centered with respect to the boundary and associated with a small duality gap. Finally, we utilize such approximate primal-dual solution pairs to warm-start an interior point solver. IPM-LSTM has several attractive features: (*i*) it can be applied to *general NLPs*; (*ii*) it strikes a *good balance between feasibility and optimality* in the returned solutions; (*iii*) it can *warm-start* and thereby accelerate interior point solvers.

The distinct contributions of our work can be summarized as follows:

- **Approximating Solutions to Linear Systems via LSTM:** This study marks the first attempt to employ learning techniques for approximating solutions of linear systems in IPMs, achieving significant speedup compared to traditional linear algebra approaches.

- **Two-Stage Framework:** We introduce a two-stage L2O framework. In the first stage, IPM-LSTM generates high-quality primal-dual solutions. In the second stage, these solutions are used to warm-start an interior point solver. This framework effectively accelerates the solving process of IPMs while yielding optimal solutions.

- **Empirical Results:** Compared with existing L2O algorithms, IPM-LSTM demonstrates favorable performance in terms of solution feasibility and optimality across various NLP types, including QPs and QCQPs. Utilizing these solutions as initial points in the state-of-the-art NLP solver IPOPT (Wächter and Biegler, 2006) reduces iterations by up to $60\%$ and solution time by up to $70\%$.

## 2 Related Works

**Constrained L2O.** Approaches utilizing L2O for constrained optimization can be broadly categorized into two directions: (i) direct learning of the mapping from optimization inputs to full solutions, and (ii) integration of learning techniques alongside or within optimization algorithms (Bengio et al., 2021; Donti et al., 2021). Previous works (Fioretto et al., 2020; Huang et al., 2021; Pan et al., 2023) adopted the former approach, employing a supervised learning scheme to train the mapping. However, this method necessitates a large number of (near-)optimal solutions as training samples, making it resource-intensive. From a self-supervised learning perspective, an intuitive approach is to incorporate the objective and penalization for constraint violation directly into the loss function (Kim et al., 2023; Park and Van Hentenryck, 2023). Nevertheless, such an approach may not guarantee the feasibility of the returned solutions. To address the feasibility issue, notable works such as Donti et al. (2021) first predict a partial solution via neural networks and then complete the full solution by utilizing equality constraints, iteratively correcting the solution towards the satisfaction of inequalities by applying gradient-based methods. However, for general nonlinear inequalities, this correction step may not ensure feasibility (Liang et al., 2023). Other approaches like Li et al. (2023) utilized gauge mappings to enforce feasibility for linear inequalities, while Liang et al. (2023) proposed the homeomorphic projection scheme to guarantee feasibility. However, these methods have limitations; the former is only applicable to linearly constrained problems, and the latter works for problems with feasibility regions homeomorphic to a unit ball. Another critical issue with the approach in Donti et al. (2021) is that the completion step may fail during training when the equality system with some fixed variables becomes infeasible, as highlighted in Han et al. (2024) and Zeng et al. (2024). Consequently, such an approach will not succeed during the training stage. To mitigate this issue, Han et al. (2024) proposed solving a projection problem if the completion step fails. However, the computationally expensive projection step may still be necessary during inference, which hinders its practical value.

**Learning-Based IPMs.** Primal-dual IPMs are polynomial-time algorithms used for solving constrained optimization problems such as LPs and NLPs. The work of Qian et al. (2024) demonstrated that properly designed *Graph Neural Networks* (GNNs) can theoretically align with IPMs for LPs, enabling GNNs to function as lightweight proxies for solving LPs. However, extending this alignment to NLPs is challenging as representation learning for general NLPs remains unknown. Another avenue of research in learning-based IPMs involves warm-starting implementation. Previous works like Baker (2019), Diehl (2019) and Zhang and Zhang (2022) addressed alternative current optimal power flow (ACOPF) applications and proposed using learning models, such as GNNs, to learn the mapping between ACOPF and its optimal solutions. These predicted solutions are then utilized as initial points to warm-start an interior point optimizer. However, even if these solutions are close to the optimal ones, they may not be well-centered with respect to the trajectory in IPMs, causing the optimizer to struggle in progressing towards feasibility and optimality (Forsgren, 2006).

## 3 Approach

### 3.1 The Classic IPM

We focus on solving the following NLP (1):

$$
\begin{aligned}
\min_{x \in \mathbb{R}^n} \quad & f(x) \\
\text{s.t.} \quad & h(x) = 0 \\
& x \geq 0
\end{aligned}
\tag{1}
$$

where the functions $f : \mathbb{R}^n \to \mathbb{R}$ and $h : \mathbb{R}^n \to \mathbb{R}^m$ are all assumed to be twice continuously differentiable. Problems with general nonlinear inequality constraints can be reformulated in the above form by introducing slack variables. We note that for simplicity, we assume all variables in (1) are non-negative, though NLPs with arbitrary variable bounds can also be handled effectively. Readers are referred to Appendix A for details.

The primal-dual IPM stands as one of the most widely utilized approaches for addressing NLPs. It entails iteratively solving the perturbed *Karush-Kuhn-Tucker* (KKT) conditions (2) for a decreasing

sequence of parameters $\mu$ converging to zero.

$$
\begin{aligned}
\nabla f(x) + \lambda^\top \nabla h(x) - z = 0 && h(x) = 0 \\
\operatorname{diag}(z)\operatorname{diag}(x)e = \mu e && x, z \geq 0
\end{aligned}
\tag{2}
$$

where $\lambda \in \mathbb{R}^m$ and $z \in \mathbb{R}_+^n$ denote the corresponding dual variables, $\operatorname{diag}(\cdot)$ represents a diagonal matrix, and $e$ is a vector of ones. Let $F(x, \lambda, z) = 0$ denote the system of nonlinear equations in (2). We then employ a one-step *Newton's method* to solve such a system, aiming to solve systems of linear equations (3).

$$
\underbrace{\begin{bmatrix}
\nabla^2 f(x) + \lambda^\top \nabla^2 h(x) & \nabla h^\top(x) & -I \\
\nabla h(x) & & \\
\operatorname{diag}(z) & & \operatorname{diag}(x)
\end{bmatrix}}_{J}
\begin{bmatrix}
\Delta x \\
\Delta \lambda \\
\Delta z
\end{bmatrix}
= -F(x, \lambda, z)
\tag{3}
$$

The IPM commences with an initial solution $(x^0, \lambda^0, z^0)$ such that $x^0, z^0 > 0$. At iteration $k$, the linear system (3) defined by the current iterate $(x^k, \lambda^k, z^k)$ is solved, with $\mu := \sigma \left[ (z^k)^\top x^k \right] / n$ being the perturbation parameter and constant $\sigma \in (0, 1)$. A line-search filter step along the direction $(\Delta x^k, \Delta \lambda^k, \Delta z^k)$ is then performed to ensure boundary condition satisfaction as well as sufficient progress towards objective value improvement or constraint violation reduction. This process iterates until convergence criteria, such as achieving optimality and feasibility within specified tolerances, are met. The IPM is guaranteed to converge to a KKT point with a superlinear rate. A pseudocode of the IPM is presented as Algorithm 1.

---

**Algorithm 1** The classic IPM

---

**Inputs:** An initial solution $(x^0, \lambda^0, z^0)$, $\sigma \in (0, 1)$, $k \leftarrow 0$
**Outputs:** The optimal solution $(x^*, \lambda^*, z^*)$
 1: **while** not converged **do**
 2:      Update $\mu^k$
 3:      Solve the system $J^k \left[ (\Delta x^k)^\top, (\Delta \lambda^k)^\top, (\Delta z^k)^\top \right]^\top = -F^k$
 4:      Choose $\alpha^k$ via a line-search filter method
 5:      $(x^{k+1}, \lambda^{k+1}, z^{k+1}) \leftarrow (x^k, \lambda^k, z^k) + \alpha^k(\Delta x^k, \Delta \lambda^k, \Delta z^k)$
 6:      $k \leftarrow k + 1$
 7: **end while**

---

We note that, in classic IPMs, one typically reformulates the system (3) and then solves a reduced system of equations (i.e., augmented system) for greater efficiency. However, in this work, we are interested in the full systems since they are associated with smaller condition numbers (Greif et al., 2014) that are critical to the performance of our proposed approach. Additionally, various techniques have been proposed to enhance the robustness and efficiency of IPMs, including second-order correction, inertial correction, and feasibility restoration. Interested readers are referred to Wächter and Biegler (2006) for further details about IPMs.

### 3.2 Approximating Solutions to Linear Systems

The small number of iterations in IPMs does not always guarantee efficiency because, at times, IPMs encounter a high per-iteration cost of linear algebra operations. In the worst-case scenario, the cost of solving a dense optimization problem using a direct linear algebra method to solve the Newton equation system (3) may reach $\mathcal{O}(n^3)$ flops per iteration. This motivates us to avoid computing exact solutions to linear systems and instead focus on their approximations. Toward this goal, we consider the following *least squares problem* (4):

$$
\min_y \frac{1}{2} \left\| J^k y + F^k \right\|^2,
\tag{4}
$$

where $\|\cdot\|$ denotes the Euclidean norm. If (3) is solvable, then an optimal solution to problem (4) is also the exact solution $\left[ (\Delta x^k)^\top, (\Delta \lambda^k)^\top, (\Delta z^k)^\top \right]^\top$ to system (3). Otherwise, we resort to an approximation of the latter. This perspective is similar to the *inexact IPM* (Bellavia, 1998; Dexter et al., 2022).

**Assumption 1.** *At iteration $k$, we could identify some $y^k$ such that*

$$\left\| J^k y^k + F^k \right\| \leq \eta \left[ (z^k)^\top x^k \right] / n \tag{5}$$

$$\| y^k \| \leq (1 + \sigma + \eta) \| F_0(x^k, \lambda^k, z^k) \|. \tag{6}$$

*where $\eta \in (0, 1)$ and $F_0(x^k, \lambda^k, z^k)$ denotes $F(x^k, \lambda^k, z^k)$ with $\mu = 0$.*

To satisfy Assumption 1, the approximate solution $y^k$ has to be bounded and accurate enough, regardless of whether $J^k$ is invertible.

**Proposition 1** (Bellavia (1998)). *If $(x^k, \lambda^k, z^k)$ is generated such that Assumption 1 is satisfied, let $(x^*, \lambda^*, z^*)$ denote a limit point of the sequence $\{(x^k, \lambda^k, z^k)\}$, then $\{(x^k, \lambda^k, z^k)\}$ converges to $(x^*, \lambda^*, z^*)$ and $F_0(x^*, \lambda^*, z^*) = 0$.*

Proposition 1 implies that if solutions with specified accuracy for linear systems in Step 3 are found, the IPM would converge.

### 3.3 The IPM-LSTM Approach

The problem (4) is an unconstrained convex optimization problem. Various L2O methods have been proposed to solve such problems (Chen et al., 2022a; Gregor and LeCun, 2010; Liu et al., 2023). We will employ the LSTM networks in our L2O method for addressing problem (4), hence our approach is called "IPM-LSTM".

**Model Architecture.** LSTM is a type of *recurrent neural network* designed to effectively capture and maintain long-term dependencies in sequential data (Yu et al., 2019). LSTM networks are commonly considered suitable for solving unconstrained optimization problems due to the resemblance between LSTM recurrent calculations and iterative algorithms (Andrychowicz et al., 2016; Liu et al., 2023; Lv et al., 2017).

The LSTM network consists of $T$ cells parameterized by the same learnable parameters $\theta$. Each cell can be viewed as one iteration of a traditional iterative method, as illustrated in Figure 2. Let $\phi(y) := \frac{1}{2} \left\| J^k y + F^k \right\|^2$ for convenience. The $t$-th cell takes the previous estimate $y_{t-1}$ and the gradient $(J^k)^\top (J^k y_{t-1} + F^k)$ as the input and outputs the current estimate $y_t$:

$$y_t := \text{LSTM}_\theta \left( \left[ y_{t-1}, (J^k)^\top (J^k y_{t-1} + F^k) \right] \right). \tag{7}$$

The $T$-th cell yields $y_T$ as an approximate solution to problem (4). As suggested by Andrychowicz et al. (2016) and Liu et al. (2023), we utilize a coordinate-wise LSTM that shares parameters not only across different LSTM cells but also for all coordinates of $y$.

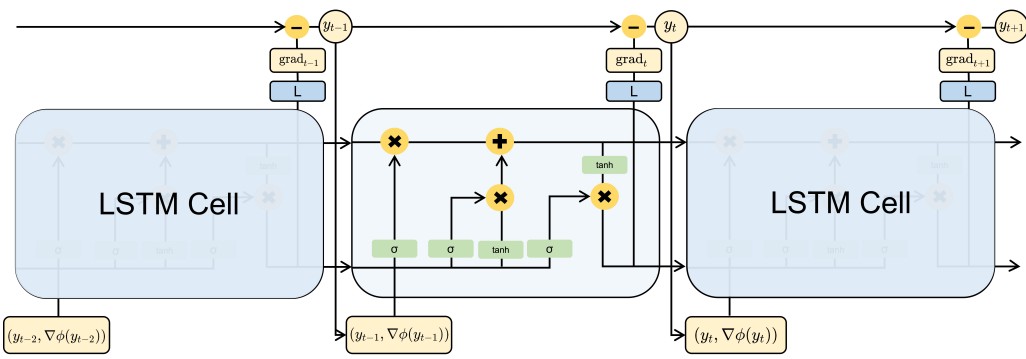

Figure 2: The LSTM architecture for solving $\min_y \phi(y)$.

**Model Training.** We train the proposed optimizer by finding the optimal $\theta$ in (7) on a dataset $\mathcal{M}$ of NLPs. Each sample in $\mathcal{M}$ is an instance of the optimization problem. During training, we apply the optimizer to each instance $M \in \mathcal{M}$, performing $K$ IPM iterations in the outer loop and $T$ LSTM

time steps in the inner loop, generating a sequence of iterates $\{(y_1^1, ..., y_T^1), ..., (y_1^K, ..., y_T^K)\}$ where the superscript $k$ denotes the IPM iteration number. We then optimize $\theta$ by minimizing the following loss function:

$$\min_\theta \frac{1}{|\mathcal{M}|} \sum_{M \in \mathcal{M}} \left( \frac{1}{K} \sum_{k=1}^K \frac{1}{T} \sum_{t=1}^T \frac{1}{2} \left\| J^k y_t^k(\theta) + F^k \right\|^2 \right)_M,$$

where the subscript $M$ indicates that the corresponding term is associated with instance $M$. Clearly, our model training falls into the category of *self-supervised learning*. To mitigate memory issues caused by excessively large computational graphs, we employ *truncated backpropagation through time* after each IPM iteration during training, as done in Chen et al. (2022a) and Liu et al. (2023).

**Preconditioning.** The Hessian matrix of $\phi(y)$ is $(J^k)^\top J^k$, whose condition number, $\kappa((J^k)^\top J^k)$, is the square of that of $J^k$. Consequently, $\kappa((J^k)^\top J^k)$ can easily become very large. Since solving system (4) via LSTM networks emulates iterative first-order methods, the value of $\kappa((J^k)^\top J^k)$ strongly affects the performance of LSTM networks. To address this issue, we employ a simple diagonal preconditioning technique that rescales the Hessian matrix $(J^k)^\top J^k$ using the Ruiz scaling method (Ruiz, 2001) to decrease its condition number.

### 3.4 Two-Stage Framework

To further enhance the solution quality, we propose a two-stage framework that initially obtains a near-optimal and well-centered primal-dual solution via IPM-LSTM and then utilizes this approximate solution to warm-start an interior point solver. In this study, we select IPOPT (Wächter and Biegler, 2006), an IPM-based solver renowned for its robustness and efficiency in optimizing NLPs.

Our two-stage framework works as follows: Given an NLP (1), we generate an initial point $(x^0, \lambda^0, z^0)$ and formulate the least squares problem (4). Subsequently, the trained LSTM network with $T$ cells solves this problem and returns a search direction. We then employ the simple fractional-to-boundary method (Wächter and Biegler, 2006) to determine the step size and reach the new iterate. This procedure is iterated $K$ times, resulting in a primal-dual solution $(x^K, \lambda^K, z^K)$. Finally, the obtained solution serves as the warm-start solution for IPOPT, leading to the optimal solution $x^*$ upon IPOPT convergence.

## 4 Experiments

### 4.1 Experimental Settings

We evaluate our approach and compare its performance against traditional methods as well as L2O algorithms for solving various types of NLPs. Furthermore, we also quantify the warm-starting effect of our proposed two-stage approach. Our code is available at https://github.com/NetSysOpt/IPM-LSTM.

**Baseline Algorithms.** In our experiments, we denote our algorithm by `IPM-LSTM` and compare it against both traditional optimizers and L2O algorithms. The traditional optimizers considered are: (i) `OSQP` (Stellato et al., 2020): an ADMM-based solver designed for convex QPs. (ii) `IPOPT` 3.14.8 (Wächter and Biegler, 2006): a state-of-the-art IPM-based solver for NLPs with the default linear solver MUMPS (Amestoy et al., 2000) and a convergence tolerance of $10^{-4}$. We also assess several L2O algorithms, including: (i) `NN` (Donti et al., 2021): a straightforward deep learning approach that integrates the objective function and penalty for constraint violations into the loss function. (ii) `DC3` (Donti et al., 2021): an end-to-end method that uses "completion" steps to maintain equality constraints and "correction" steps for inequality feasibility. (iii) `DeepLDE` (Kim et al., 2023): an algorithm that trains neural networks using a primal-dual approach to impose inequality constraints and employs "completion" for equality constraints. (iv) `PDL` (Park and Van Hentenryck, 2023): a self-supervised learning method that jointly trains two networks to approximate primal and dual solutions. (v) `LOOP-LC` (Li et al., 2023): a neural approximator that maps inputs of linearly constrained models to high-quality feasible solutions using gauge maps. (vi) `H-Proj` (Liang et al., 2023): a method that applies a homeomorphic projection scheme to post-process solutions resulting from the completion step in `DC3`.

**Datasets.** The dataset used in this paper includes randomly generated benchmarks obtained from Chen and Burer (2012), Donti et al. (2021) and Liang et al. (2023), as well as real-world instances

from Globallib (see http://www.minlplib.org). These benchmarks encompass QPs, QCQPs, and simplex non-convex programs. For each case, we generate $10,000$ samples and divide them into a $10:1:1$ ratio for training, validation, and testing, respectively. All numerical results are reported for the test set.

**Model Settings.** All LSTM networks have a single layer and are trained using the Adam optimizer (Kingma, 2014). During `IPM-LSTM` training, an early stopping strategy with a patience of $50$ is employed, halting training if no improvement is observed for $50$ iterations, while satisfying inequality and equality constraints violation less than $0.005$ and $0.01$. The learning rate is $0.0001$, and the batch size is $128$ for each task. Additional `IPM-LSTM` parameters for each task are provided in Appendix C.

**Evaluation Configuration.** All our experiments were conducted on an NVIDIA RTX A6000 GPU, an Intel Xeon 2.10GHz CPU, using Python 3.10.0 and PyTorch 1.13.1.

## 4.2 Computational Results

**Convex QPs.** We consider convex QPs with both equality and inequality constraints:

$$
\begin{aligned}
\min_{x \in \mathbb{R}^n} \quad & \frac{1}{2} x^\top Q_0 x + p_0^\top x \\
\text{s.t.} \quad & p_j^\top x \le q_j & j = 1, \cdots, l \\
& p_j^\top x = q_j & j = l+1, \cdots, m \\
& x_i^L \le x_i \le x_i^U & i = 1, \cdots, n
\end{aligned}
\tag{8}
$$

where $Q_0 \in \mathbb{S}_+^n$, $p_j \in \mathbb{R}^n$, $q_j \in \mathbb{R}$, $x_i^L \in \mathbb{R} \cup \{-\infty\}$ and $x_i^U \in \mathbb{R} \cup \{+\infty\}$. We conduct experiments on two groups of QPs, each instance with 200 variables, 100 inequalities and 100 equalities. The first group is generated in the same way as Donti et al. (2021), where only the right hand sides of equality constraints are perturbed while the second one considers perturbation for all model parameters. Let "Convex QP (RHS)" denote the former and "Convex QPs (ALL)" denote the latter. It is noteworthy that, in line with Donti et al. (2021), we also investigate the performance of `IPM-LSTM` and baseline algorithms on smaller-scale convex QPs. Interested readers are directed to Appendix D for details.

Table 1: Computational results on convex QPs.

| Method | End-to-End | | | | | | IPOPT (warm start) | | Total Time (s) ↓ | Gain (Ite./ Time) ↑ |
|---|---|---|---|---|---|---|---|---|---|---|
| | Obj. ↓ | Max ineq. ↓ | Mean ineq. ↓ | Max eq. ↓ | Mean eq. ↓ | Time (s) ↓ | Ite. ↓ | Time (s) ↓ | | |
| **Convex QPs (RHS)** | | | | | | | | | | |
| OSQP | -29.176 | 0.000 | 0.000 | 0.000 | 0.000 | 0.009 | - | - | - | - |
| IPOPT | -29.176 | 0.000 | 0.000 | 0.000 | 0.000 | 0.642 | 12.5 | - | - | - |
| NN | -26.787 | 0.000 | 0.000 | 0.631 | 0.235 | <0.001 | 10.5 | 0.560 | 0.560 | 16.0%/12.8% |
| DC3 | -26.720 | 0.002 | 0.000 | 0.000 | 0.000 | <0.001 | 10.2 | 0.535 | 0.535 | 18.4%/16.7% |
| DeepLDE | -3.697 | 0.000 | 0.000 | 0.000 | 0.000 | <0.001 | 12.5 | 0.648 | 0.648 | 0.0%/-0.9% |
| PDL | -28.559 | 0.421 | 0.122 | 0.024 | 0.000 | <0.001 | 9.7 | 0.514 | 0.514 | 22.4%/**19.9%** |
| LOOP-LC | -28.512 | 0.000 | 0.000 | 0.000 | 0.000 | <0.001 | 10.8 | 0.565 | 0.565 | 13.6%/12.0% |
| H-Proj | -23.257 | 0.000 | 0.000 | 0.000 | 0.000 | <0.001 | 11.2 | 0.605 | 0.605 | 10.4%/5.8% |
| IPM-LSTM | -29.050 | 0.000 | 0.000 | 0.002 | 0.001 | 0.175 | 7.2 | 0.370 | 0.545 | **42.4%**/15.1% |
| **Convex QPs (ALL)** | | | | | | | | | | |
| OSQP | -33.183 | 0.000 | 0.000 | 0.000 | 0.000 | 0.009 | - | - | - | - |
| IPOPT | -33.183 | 0.000 | 0.000 | 0.000 | 0.000 | 0.671 | 12.9 | - | - | - |
| IPM-LSTM | -32.600 | 0.000 | 0.000 | 0.003 | 0.001 | 0.195 | 8.3 | 0.426 | 0.621 | **35.7%/7.5%** |

We ran `IPM-LSTM` and all baseline algorithms on the test set "Convex QPs (RHS)", and we reported their computational results, which were averaged across 833 instances. The results are presented in Table 1. We denote this experiment by "End-to-End" for convenience. The columns labeled "Max Ineq.", "Mean Ineq.", "Max Eq.", and "Mean Eq." denote the maximum and mean violations for inequalities and equalities, respectively. The columns "Obj." and "Time (s)" represent the final primal objective and the runtime in seconds. Both `OSQP` and `IPOPT` solved these instances to guaranteed optimality, with `OSQP` being significantly more efficient due to its specialization as a QP-specific solver. All L2O baseline algorithms returned solutions very quickly. However, solutions generated by `NN` and `PDL` exhibited significant constraint violations. While the solutions from `DC3` and `DeepLDE` were nearly feasible, they corresponded to inferior objective values. On the other hand, `LOOP-LC` produced feasible and near-optimal solutions, whereas `H-Proj` generated feasible solutions but with

larger objective values. Solutions identified by `IPM-LSTM` showed mild constraint violations but yielded superior objective values very close to the optimal values. Clearly, `IPM-LSTM` effectively balances feasibility and optimality in the returned solutions. This comes at the cost of longer runtime compared to the L2O baseline algorithms, as the baselines typically employ simple multi-layer perceptrons, whereas `IPM-LSTM` utilizes a neural network with several dozen LSTM cells.

Since `IPM-LSTM` is designed to provide interior point optimizers with high-quality initial points, we fed the returned primal-dual solution pair to `IPOPT` and reported the performance in Table 1. For comparison, we also provided `IPOPT` with initial points generated from other L2O algorithms. The columns labeled "Ite." and "Time (s)" under "`IPOPT` (warm-start)" indicate the number of iterations and solver time, respectively, while the column "Total Time (s)" represents the cumulative time for running both the L2O algorithms and `IPOPT`. The final column, "Gain (Ite/Time)", shows the reduction in iteration number and solution time, with the default `IPOPT` iteration number listed in the "Ite." column for reference. When initial solutions from `IPM-LSTM` and most L2O baseline algorithms (except `DeepLDE`) were used, `IPOPT` converged with fewer iterations and reduced solution time. Notably, `IPM-LSTM` achieved the most significant reduction in iterations, from $12.5$ to $7.2$, and decreased the average solver time from $0.642$ seconds to $0.37$ seconds. Including the computational time for `IPM-LSTM`, the total runtime was $0.545$ seconds, reflecting a $15.1\%$ reduction in time. It is worth noting that while `IPM-LSTM` did not yield the maximum solution time reduction, this was due to its relatively high computational expense.

To our knowledge, there is no existing representation learning approach for general convex QPs. Since the aforementioned L2O baseline algorithms depend on specific representations of QPs, they are not applicable to "Convex QPs (ALL)". Therefore, we only provide results for `OSQP`, `IPOPT`, and `IPM-LSTM`. We also report computational results averaged across $833$ instances in the test set "Convex QPs (ALL)", presented in Table 1. The results demonstrate that `IPM-LSTM` can identify high-quality solutions for general convex QPs, and using these solutions as initial points can reduce iterations by $35.7\%$ and solution time by $7.5\%$.

**Convex QCQPs.** We now turn to convex QCQPs with both equaltity and inequality constraints:

$$
\begin{aligned}
\min_{x \in \mathbb{R}^n} \quad & \frac{1}{2} x^\top Q_0 x + p_0^\top x \\
\text{s.t.} \quad & x^\top Q_j x + p_j^\top x \le q_j && j = 1, \cdots, l \\
& p_j^\top x = q_j && j = l+1, \cdots, m \\
& x_i^L \le x_i \le x_i^U && i = 1, \cdots, n
\end{aligned}
$$

where $Q_j \in \mathbb{S}_+^n$, $p_j \in \mathbb{R}^n$, $q_j \in \mathbb{R}$, $x_i^L \in \mathbb{R} \cup \{-\infty\}$ and $x_i^U \in \mathbb{R} \cup \{\infty\}$. Similar to our experiments on convex QPs, we also consider two groups of convex QCQPs, each with $200$ variables, $100$ inequality constraints, and $100$ equality constraints. The first group (denoted as "Convex QCQPs (RHS)") is generated as described in Liang et al. (2023), with perturbations only to the right-hand sides of the equality constraints. The second group (denoted as "Convex QCQPs (ALL)") considers perturbations to all parameters. We also refer readers to Appendix D for computational experiments on smaller-sized convex QCQPs.

We omit `OSQP` and `LOOP-LC` since the former cannot handle QCQPs, while the latter is only applicable to linearly constrained problems. We evaluate `IPM-LSTM` and compare it against the remaining baseline algorithms. The computational results are reported in Table 2. Again, solutions produced by `NN` and `PDL` exhibit significant constraint violations, while those from `DC3` and `H-Proj` are of high quality in terms of feasibility and optimality. Once more, the solutions produced by `DeepLDE` were deemed feasible but exhibited inferior objective values. Conversely, our approach, `IPM-LSTM`, produced solutions with superior objective values albeit with minor infeasibility. Compared to the baseline algorithms, utilizing solutions from `IPM-LSTM` to warm-start `IPOPT` resulted in the most substantial reduction in iterations.

As the aforementioned L2O baseline algorithms are not applicable to "Convex QCQPs (ALL)", we only report computational results for `IPOPT` and `IPM-LSTM` in Table 2. The `IPM-LSTM` approach produced high-quality approximate solutions to convex QCQPs, and warm-starting `IPOPT` with these solutions accelerated `IPOPT` by $11.4\%$, with a $33.1\%$ reduction in iterations.

**Non-convex QPs.** We now consider non-convex QPs of exactly the same form as (8) but with $Q_0$ being indefinite. We take $8$ representative non-convex QPs from the datasets Globallib and

Table 2: Computational results on convex QCQPs.

| Method | End-to-End | | | | | | IPOPT (warm start) | | Total Time (s)↓ | Gain (Ite./ Time)↑ |
|---|---|---|---|---|---|---|---|---|---|---|
| | Obj. ↓ | Max ineq. ↓ | Mean ineq. ↓ | Max eq. ↓ | Mean eq. ↓ | Time (s) ↓ | Ite. ↓ | Time (s) ↓ | | |
| **Convex QCQPs (RHS)** | | | | | | | | | | |
| IPOPT | -39.162 | 0.000 | 0.000 | 0.000 | 0.000 | 1.098 | 12.5 | - | - | - |
| NN | -2.105 | 0.000 | 0.000 | 0.552 | 0.169 | <0.001 | 12.1 | 1.311 | 1.311 | 3.2%/-19.4% |
| DC3 | -35.741 | 0.000 | 0.000 | 0.000 | 0.000 | 0.005 | 9.6 | 1.051 | 1.051 | 20.7%/4.8% |
| DeepLDE | -15.132 | 0.000 | 0.000 | 0.000 | 0.000 | <0.001 | 11.5 | 1.222 | 1.222 | 8.0%/-11.3% |
| PDL | -39.089 | 0.005 | 0.000 | 0.015 | 0.005 | <0.001 | 8.9 | 1.013 | 1.013 | 28.8%/**7.7%** |
| H-Proj | -36.062 | 0.000 | 0.000 | 0.000 | 0.000 | <0.001 | 9.8 | 1.070 | 1.070 | 21.6%/2.6% |
| IPM-LSTM | -38.540 | 0.000 | 0.000 | 0.004 | 0.001 | 0.205 | 8.0 | 0.825 | 1.030 | **36.0%**/6.2% |
| **Convex QCQPs (ALL)** | | | | | | | | | | |
| IPOPT | -39.868 | 0.000 | 0.000 | 0.000 | 0.000 | 0.801 | 12.4 | - | - | - |
| IPM-LSTM | -38.405 | 0.004 | 0.000 | 0.001 | 0.000 | 0.203 | 8.3 | 0.507 | 0.710 | **33.1%/11.4%** |

RandQP (Chen and Burer, 2012), each with up to 50 variables and 20 constraints, and perturb the relevant parameters for instance generation. Details can be found in Appendix D.2.

Among all the baselines, only IPOPT is applicable to these general non-convex QPs. Hence, we report computational results for IPOPT and IPM-LSTM in Table 3. IPOPT solved these instances to local optimality, while IPM-LSTM identified high-quality approximate solutions very efficiently. Using these primal-dual approximations to warm-start IPOPT resulted in an iteration reduction of up to 63.9% and a solution time reduction of up to 70.5%.

Table 3: Computational results on non-convex QPs.

| Instance | IPOPT | | | IPM-LSTM | | | IPOPT (warm-start) | | | Total Time (s) | Gain (Ite./ Time) |
|---|---|---|---|---|---|---|---|---|---|---|---|
| | Obj. | Ite. | Time (s) | Obj. | Max Vio. | Time (s) | Obj. | Ite. | Time (s) | | |
| qp1 | 0.001 | 52.0 | 0.707 | 0.045 | 0.008 | 0.017 | 0.001 | 42.0 | 0.559 | 0.576 | 19.2%/18.5% |
| qp2 | 0.001 | 69.0 | 0.674 | 0.034 | 0.008 | 0.029 | 0.001 | 40.0 | 0.347 | 0.376 | 42.0%/ 44.2% |
| st_rv1 | -58.430 | 215.0 | 0.955 | -34.563 | 0.000 | 0.009 | -58.867 | 168.0 | 0.626 | 0.635 | 21.9%/33.5% |
| st_rv2 | -67.083 | 190.8 | 0.956 | -30.955 | 0.000 | 0.011 | -67.083 | 120.5 | 0.482 | 0.494 | 36.8%/38.1% |
| st_rv3 | 0.000 | 55.0 | 0.781 | 0.818 | 0.000 | 0.017 | 0.000 | 47.0 | 0.616 | 0.634 | 14.5%/18.8% |
| st_rv7 | -132.019 | 449.0 | 2.445 | -61.428 | 0.000 | 0.016 | -131.756 | 162.0 | 0.705 | 0.721 | 63.9%/70.5% |
| st_rv9 | -126.945 | 655.0 | 3.457 | -58.415 | 0.000 | 0.026 | -127.652 | 408.0 | 1.830 | 1.856 | 37.7%/46.3% |
| qp30_15_1_1 | 37.767 | 16.0 | 0.198 | 37.787 | 0.002 | 0.021 | 37.767 | 9.0 | 0.083 | 0.104 | 43.7%/47.5% |

Max Vio. denotes the maximum constraint violation.

**Simple non-convex programs.** Following the approach outlined in Donti et al. (2021), we consider a set of simple non-convex programs where the linear objective term in (8) is substituted with $p_0^\top \sin(x)$. These instances were generated using the same methodology as Donti et al. (2021). We subject these problems to evaluation using both IPM-LSTM and baseline algorithms. Once again, the computational results underscore the high-quality solutions obtained by IPM-LSTM and its superior warm-starting capability. Detailed results are provided in Appendix D.4.

### 4.3 Performance Analysis of IPM-LSTM

In Assumption 1, we posit that the linear systems (3) can be solved with an acceptable residual (Condition (5)) and that the solutions are properly bounded (Condition (6)). Although the LSTM network cannot guarantee the satisfaction of these conditions, we empirically assess the validity of Assumption 1 in IPM-LSTM. We plot the progress of $\left\| J^k y^k + F^k \right\|, \eta \left[ (z^k)^\top x^k \right] / n, \left\| y^k \right\|$, and $\left\| F_0(x^k, \lambda^k, z^k) \right\|$ as the IPM iterations increase in Figure 3(a) and 3(b). Condition (5) is mostly satisfied except during the first few iterations, while Condition (6) is strictly satisfied across all IPM iterations. To assess the precision of the approximate solutions as the LSTM time steps increase, we plot the progress of the residual for linear systems (3) in Figure 3(c). At IPM iteration $k$, the residual decreases monotonically towards 0 as the LSTM time steps increase, indicating that LSTM networks can produce high-quality approximate solutions to system (3). The approximation quality improves with the number of IPM iterations. As shown in Figure 3(d), the primal objective decreases monotonically towards the optimal value with increasing IPM iterations, empirically demonstrating the convergence of IPM-LSTM. Note that the condition number $\kappa((J^k)^\top J^k)$ becomes quite large in the later IPM iterations, which can adversely affect the performance of LSTM networks in obtaining approximations to systems (3). Consequently, we terminate IPM-LSTM after a finite number of iterations. Further analysis regarding the number of IPM iterations and LSTM time steps is

presented in Appendix D.5. Additionally, we include the performance of `IPM-LSTM` under various hyperparameter settings in Appendix D.5.

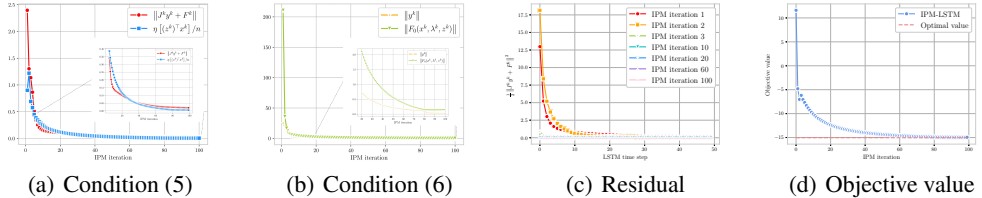

| (a) Condition (5) | (b) Condition (6) | (c) Residual | (d) Objective value |

Figure 3: The performance analysis of IPM-LSTM on a convex QP (RHS).

## 5 Limitations and Conclusions

In this paper, we present a learning-based IPM called IPM-LSTM. Specifically, we propose approximating solutions of linear systems in IPMs by solving least square problems using trained LSTM networks. We demonstrate that IPMs with this approximation procedure still converge. The solutions returned by IPM-LSTM are used to warm-start interior point optimizers. Our computational experiments on various types of NLPs, including general QPs and QCQPs, showcase the effectiveness of IPM-LSTM and its ability to accelerate IPOPT. Although IPM-LSTM generates high-quality primal-dual solutions, it is relatively computationally expensive due to the utilization of multi-cell LSTM networks. In future endeavors, we aim to investigate the efficacy of employing low-complexity neural networks to approximate solutions of linear systems within IPMs.

## Acknowledgments

This work was supported by the National Key R&D Program of China under grant 2022YFA1003900. Jinxin Xiong and Akang Wang also gratefully acknowledge support from the National Natural Science Foundation of China (Grant No. 12301416), the Shenzhen Science and Technology Program (Grant No. RCBS20221008093309021), the Guangdong Basic and Applied Basic Research Foundation (Grant No. 2024A1515010306) and Longgang District Special Funds for Science and Technology Innovation (LGKCSDPT2023002).

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

## Appendix

## A   Implementation Details

The general NLP considered in this paper can be formulated as:

$$
\begin{aligned}
\min_{x \in \mathbb{R}^n} \quad & f(x), \\
\text{s.t.} \quad & g(x) + s = 0 \\
& h(x) = 0 \\
& s \geq 0 \\
& x_i \geq x_i^L, \qquad i \in I^L \\
& x_i \leq x_i^U, \qquad i \in I^U
\end{aligned}
\tag{9}
$$

where $f : \mathbb{R}^n \to \mathbb{R}$, $g : \mathbb{R}^n \to \mathbb{R}^{m_{\text{ineq}}}$, and $h : \mathbb{R}^n \to \mathbb{R}^{m_{\text{eq}}}$ are twice continuously differentiable functions; $s \in \mathbb{R}^{m_{\text{ineq}}}$ is the slack variable corresponding to the inequality constraints; $I^L = \{i : x_i^L \neq -\infty\}$ and $I^U = \{i : x_i^U \neq \infty\}$. The Lagrangian function is defined as:

$$
L\left(x, \eta, \lambda, s, z^L, z^U\right) := f(x) + \eta^\top g(x) + \lambda^\top h(x) - \sum_{i \in I^L} z_i^L \left(x_i - x_i^L\right) - \sum_{i \in I^U} z_i^U \left(x_i^U - x_i\right)
\tag{10}
$$

where $\eta \in \mathbb{R}^{m_{\text{ineq}}}$ and $\lambda \in \mathbb{R}^{m_{\text{eq}}}$ are the corresponding dual variables. The perturbed KKT conditions are given by:

$$
\begin{cases}
\nabla f(x) + \eta^\top \nabla g(x) + \lambda^\top \nabla h(x) - z^L + z^U = 0 \\
g(x) + s = 0 \\
\text{diag}(\eta)\text{diag}(s)e = \mu e \\
\eta \geq 0, s \geq 0, h(x) = 0 \\
z_i^L(x_i - x_i^L) = \mu, i \in I^L \\
z_i^U(x_i^U - x_i) = \mu, i \in I^U \\
z_i^L \geq 0, i \in I^L, z_i^U \geq 0, i \in I^U \\
x_i \geq x_i^L, i \in I^L, x_j \leq x_j^U, j \in I^U
\end{cases}
\tag{11}
$$

The nonlinear system extracted from the KKT conditions is represented as:

$$
F(x, \eta, \lambda, s, z^L, z^U) = 0
\tag{12}
$$

where

$$
F(x, \eta, \lambda, s, z^L, z^U) =
\begin{pmatrix}
\nabla f(x) + \eta^\top \nabla g(x) + \lambda^\top \nabla h(x) - z^L + z^U \\
g(x) + s \\
\text{diag}(\eta)\text{diag}(s)e - \mu e \\
h(x) \\
\text{diag}(z^L)\text{diag}(x - x^L)e - \mu e \\
\text{diag}(z^U)\text{diag}(x^U - x)e - \mu e
\end{pmatrix}
= 0
$$

The Jacobian matrix of $F(x, \eta, \lambda, s, z^L, z^U)$ is:

$$
J(x, \eta, \lambda, s, z^L, z^U) :=
\begin{pmatrix}
\nabla^2 L(x, \eta, \lambda, z^L, z^U), & \nabla g(x)^\top, & \nabla h(x)^\top, & 0, & -I, & I \\
\nabla g(x), & 0, & 0, & I, & 0, & 0 \\
0, & \text{diag}(s), & 0, & \text{diag}(\eta), & 0, & 0 \\
\nabla h(x), & 0, & 0, & 0, & 0, & 0 \\
\text{diag}(z^L), & 0, & 0, & 0, & \text{diag}(x - x^L), & 0 \\
-\text{diag}(z^U), & 0, & 0, & 0, & 0, & \text{diag}(x^U - x)
\end{pmatrix}
$$

where

$$
\nabla^2 L\left(x, \eta, \lambda, s, z^L, z^U\right) = \nabla^2 f(x) + \sum_{i=1}^{m_{\text{ineq}}} \eta_i \nabla^2 g_i(x) + \sum_{j=1}^{m_{\text{eq}}} \lambda_j \nabla^2 h_j(x)
\tag{13}
$$

For convenience, we define the updates of primal and dual variables as:

$$
y :=
\begin{pmatrix}
\Delta x \\
\Delta \eta \\
\Delta \lambda \\
\Delta s \\
\Delta z^L \\
\Delta z^U
\end{pmatrix}.
\tag{14}
$$

Then the linear system obtained from one-step Newton's method is shown as:

$$J\left(x, \eta, \lambda, s, z^L, z^U\right) y = -F\left(x, \eta, \lambda, s, z^L, z^U\right) \tag{15}$$

**Initial points.** If dual variables $\eta$, $s$, $z^L$, and $z^U$ exist, their initial values are set to 1. If the dual variable $\lambda$ exists, its initial value is set to 0. The initial value of the primal variable $x$ can be formulated as:

$$x_i = \begin{cases} z_i^L + 1, & i \in I^L \backslash I^U \\ z_i^U - 1, & i \in I^U \backslash I^L \\ (z_i^L + z_i^U)/2, & i \in I^L \cap I^U \\ 0, & \text{otherwise.} \end{cases} \tag{16}$$

**The step length.** If the dual variables $\eta$, $s$, $z^L$ and $z^U$ exist, their step lengths are chosen as follows:

$$\alpha^\eta := \sup\left\{\alpha \in (0,1] \mid \eta_i + \alpha\Delta\eta_i \geq 0, i = 1, \cdots, m_{\text{ineq}}\right\} \tag{17}$$

$$\alpha^s := \sup\left\{\alpha \in (0,1] \mid s_i + \alpha\Delta s_i \geq 0, i = 1, \cdots, m_{\text{ineq}}\right\} \tag{18}$$

$$\alpha^{z^L} := \sup\left\{\alpha \in (0,1] \mid z_i^L + \alpha\Delta z_i^L \geq 0, i \in I^L\right\} \tag{19}$$

$$\alpha^{z^U} := \sup\left\{\alpha \in (0,1] \mid z_i^U + \alpha\Delta z_i^U \geq 0, i \in I^U\right\} \tag{20}$$

If $x$ is bounded, the step sizes for $x$ and $\lambda$ are chosen to be equal as:

$$\alpha^\lambda, \alpha^x := \sup\left\{\alpha \in (0,1] \mid x_i + \alpha\Delta x_i \geq x_i^L, i \in I^L; x_i + \alpha\Delta x_i \leq x_i^U, i \in I^U\right\} \tag{21}$$

If $x$ is unbounded but inequality constraints exist, the step lengths for $x$ and $\lambda$ are chosen the same as the step length for $s$. If $x$ is unbounded and there are no inequality constraints, the step lengths for $x$ and $\lambda$ are chosen to be 1. In our experiments, additional line search procedures for step lengths, as described in Bellavia (1998), were not implemented in order to save computational time.

## B  Proof of Proposition 1

*Proof.* **Step 1.** Let $\delta > 0$, suppose that $(x^k, \lambda^k, z^k) \in N_\delta(x^*, \lambda^*, z^*) := \{(x, \lambda, z) \mid \|(x, \lambda, z) - (x^*, \lambda^*, z^*)\| < \delta\}$. Define $\sigma := \min\{\sigma^k\}$ and $\eta := \min\{\eta^k\}$, where $\sigma^k$ and $\eta^k$ are chosen to satisfy the update rule (11) in Bellavia (1998). Then from the Assumption 1:

$$\|y^k\| \leq (1 + \sigma^k + \eta^k)\|F_0(x^k, \lambda^k, z^k)\|. \tag{22}$$

By using the steplength $\alpha^k$ obtained from the line search method of Bellavia (1998) and defining:

$$\hat{\eta}^k = 1 - \alpha^k\left(1 - \sigma^k - \eta^k\right) \tag{23}$$

we obtain:

$$\begin{aligned} \|p^k\| &= \|\alpha^k y^k\| \\ &= \|\frac{(1-\hat{\eta}^k)\alpha^k y^k}{1-(1-\alpha^k(1-\sigma^k-\eta^k))}\| \\ &= \frac{1-\hat{\eta}^k}{1-\sigma^k-\eta^k}\|y^k\| \\ &\leq (1-\hat{\eta}^k)\frac{1+\sigma^k+\eta^k}{1-\sigma^k-\eta^k}\|F_0(x^k, \lambda^k, z^k)\| \\ &\leq (1-\hat{\eta}^k)\Gamma\|F_0(x^k, \lambda^k, z^k)\| \end{aligned} \tag{24}$$

where $\Gamma = \frac{1+\eta_{\max}}{1-\eta_{\max}}$, $(\sigma^k + \eta^k) \in (0, \eta_{\max})$ and $\eta_{\max} \in (0,1)$. Then, there exists a constant $\Gamma$ independent of $k$ such that (24) holds, whenever $y^k$ is bounded by $(1 + \sigma^k + \eta^k)\|F_0(x^k, \lambda^k, z^k)\|$. Hence, from Theorem 3.5 of Eisenstat and Walker (1994), it follows that $(x^k, \lambda^k, z^k) \to (x^*, \lambda^*, z^*)$.

**Step 2.** In this step, we aim to prove that the step size $\alpha^k$ of the inexact IPM is bounded away from 0. Given that Assumption 1 is satisfied, it follows that $\|J^k y^k + F^k\|$ and $\|y^k\|$ are both bounded. Then we assume the first two equations of $F_0(x, \lambda, z)$ are Lipschitz continuous gradient with constant $L$. The remaining proofs are consistent with Theorem 3.2 in Bellavia (1998).

**Step 3.** The line search method of Bellavia (1998) enables $\alpha^k$ to satisfy:

$$\|F_0(x^{k+1}, \lambda^{k+1}, z^{k+1})\| \leq (1 - \beta(1 - \hat{\eta}^k))\|F_0(x^k, \lambda^k, z^k)\| \tag{25}$$

where $\beta \in (0, 1)$. Therefore $\{\|F_0(x^k, \lambda^k, z^k)\|\}$ is decreasing and bounded, hence, it is convergent. Based on Assumption 1, (24) holds. Furthermore, we assume $\|F_0(x^k, \lambda^k, z^k)\| \neq 0$ and $\delta$ is chosen sufficiently small so that

$$
\begin{aligned}
&\|F_0(x_2, \lambda_2, z_2) - F_0(x_1, \lambda_1, z_1) - J(x_1, \lambda_1, z_1)[(x_2 - x_1)^\top, (\lambda_2 - \lambda_1)\top, (z_2 - z_1)^\top]^\top\| \\
&\quad \leq ((1 - \beta)/\Gamma)\|[(x_2 - x_1)^\top, (\lambda_2 - \lambda_1)^\top, (z_2 - z_1)^\top]^\top\|
\end{aligned}
\tag{26}
$$

is satisfied, whenever $(x_1, \lambda_1, z_1), (x_2, \lambda_2, z_2) \in N_{2\delta}(x^*, \lambda^*, z^*)$. Define

$$
S := \sup_{(x, \lambda, z) \in N_\delta(x^*, \lambda^*, z^*)} \|F_0(x, \lambda, z)\|,
\tag{27}
$$

then based on Lemma 5.1 of Eisenstat and Walker (1994), the loop of backtracking line search of Bellavia (1998) will terminate with

$$
1 - \hat{\eta}^k \geq \min(\alpha^k(1 - \sigma^k - \eta^k), \theta\delta/(\Gamma S))
\tag{28}
$$

where $\theta \in (0, 1)$ is a control parameter. This implies that the series $\sum_{k=0}^{\infty}(1 - \hat{\eta}^k)$ is divergent. By applying (25) iteratively, we have

$$
\begin{aligned}
\|F_0(x^k, \lambda^k, z^k)\| &\leq (1 - \beta(1 - \hat{\eta}^{k-1}))\|F_0(x^{k-1}, \lambda^{k-1}, z^{k-1})\| \\
&\leq \|F_0(x^0, \lambda^0, z^0)\| \prod_{0 \leq j < k}(1 - \beta(1 - \hat{\eta}^j)) \\
&\leq \|F_0(x^0, \lambda^0, z^0)\|\exp(-\beta \sum_{0 \leq j < k}(1 - \hat{\eta}^j)).
\end{aligned}
\tag{29}
$$

Since $\beta > 0$ and $1 - \hat{\eta}^j \geq 0$, the divergence of series $\sum_{j=0}^{\infty}(1 - \hat{\eta}^j)$ implies $\|F_0(x^k, \lambda^k, z^k)\| \to 0$. $\square$

## C  Datasets and Parameter setting

The key hyperparameters for each task, including $K, T$, and the hidden dimension of LSTM networks, are listed in Table 4. Generally, IPM-LSTM demonstrates improved performance with larger values of $K$ and $T$, although this comes at the cost of increased computational time. Increasing $T$ could enhance the quality of solutions to the linear system. In this study, $K$ is maintained at the same value for both training and testing. Further analysis can be found in Appendix D.5.

Table 4: Instance information and hyperparameter settings.

| Instance | Information | | | | | | Hyperparameters | | |
|---|---|---|---|---|---|---|---|---|---|
| | Source | $n$ | $m_{\text{ineq}}$ | $m_{\text{eq}}$ | $|I^L|$ | $|I^U|$ | $K$ | $T$ | Hidden dimension |
| Convex QPs (RHS) | | 100 | 50 | 50 | 0 | 0 | 100 | 50 | 50 |
| | | 200 | 100 | 100 | 0 | 0 | 100 | 50 | 75 |
| Convex QPs (ALL) | | 100 | 50 | 50 | 0 | 0 | 100 | 50 | 50 |
| | | 200 | 100 | 100 | 0 | 0 | 100 | 50 | 100 |
| Convex QCQPs (RHS) | | 100 | 50 | 50 | 0 | 0 | 100 | 50 | 50 |
| | | 200 | 100 | 100 | 0 | 0 | 100 | 50 | 100 |
| | Synthetic | 100 | 50 | 50 | 0 | 0 | 100 | 50 | 50 |
| Convex QCQPs (ALL) | | 200 | 100 | 100 | 0 | 0 | 100 | 50 | 100 |
| Non-convex Programs (RHS) | | 100 | 50 | 50 | 0 | 0 | 100 | 50 | 50 |
| | | 200 | 100 | 100 | 0 | 0 | 100 | 50 | 75 |
| Non-convex Programs (ALL) | | 100 | 50 | 50 | 0 | 0 | 100 | 50 | 50 |
| | | 200 | 100 | 100 | 0 | 0 | 100 | 50 | 100 |
| qp1 | | 50 | 1 | 1 | 50 | 0 | 100 | 50 | 30 |
| qp2 | | 50 | 1 | 1 | 50 | 0 | 100 | 50 | 30 |
| st_rv1 | | 10 | 5 | 0 | 10 | 0 | 100 | 50 | 50 |
| st_rv2 | Globallib | 20 | 10 | 0 | 20 | 0 | 100 | 50 | 50 |
| st_rv3 | | 50 | 1 | 0 | 50 | 0 | 100 | 50 | 50 |
| st_rv7 | | 30 | 20 | 0 | 30 | 0 | 100 | 50 | 50 |
| st_rv9 | | 50 | 20 | 0 | 50 | 0 | 100 | 50 | 50 |
| qp30_15_1_1 | Rand_QP | 30 | 15 | 6 | 30 | 30 | 100 | 50 | 50 |

# D   Experimental Results

## D.1   Convex QPs

The performance on convex QPs, including "Convex QPs (RHS)" and "Convex QPs (ALL)", each instance with 100 variables, 50 inequality constraints, and 50 equality constraints, is shown in Table 5. From Table 5, IPM-LSTM provided the best objective value with acceptable constraint violations.

Table 5: Computational results on convex QPs

| Method | End-to-End | | | | | | IPOPT (warm start) | | Total Time (s)↓ | Gain (Ite./ Time)↑ |
|---|---|---|---|---|---|---|---|---|---|---|
| | Obj. ↓ | Max ineq. ↓ | Mean ineq. ↓ | Max eq. ↓ | Mean eq. ↓ | Time (s) ↓ | Ite. ↓ | Time (s) ↓ | | |
| **Convex QPs (RHS)** | | | | | | | | | | |
| OSQP | -15.047 | 0.000 | 0.000 | 0.000 | 0.000 | 0.002 | - | - | - | - |
| IPOPT | -15.047 | 0.000 | 0.000 | 0.000 | 0.000 | 0.269 | 12.2 | - | - | - |
| DC3 | -13.460 | 0.000 | 0.000 | 0.000 | 0.000 | <0.001 | 10.5 | 0.227 | 0.227 | 13.9%/15.6% |
| NN | -12.570 | 0.000 | 0.000 | 0.350 | 0.130 | <0.001 | 10.7 | 0.234 | 0.234 | 12.3%/13.0% |
| DeepLDE | 46.316 | 0.000 | 0.000 | 0.007 | 0.000 | <0.001 | 12.9 | 0.294 | 0.294 | -5.7%/-9.3% |
| PDL | -14.969 | 0.011 | 0.003 | 0.002 | 0.000 | <0.001 | 9.5 | 0.199 | 0.199 | 22.1%/26.0% |
| LOOP-LC | -13.628 | 0.000 | 0.000 | 0.000 | 0.000 | <0.001 | 11.2 | 0.246 | 0.246 | 8.2%/8.6% |
| H-Proj | -11.778 | 0.000 | 0.000 | 0.000 | 0.000 | <0.001 | 10.7 | 0.233 | 0.233 | 12.3%/13.4% |
| IPM-LSTM | -14.985 | 0.000 | 0.000 | 0.001 | 0.000 | 0.045 | 6.5 | 0.115 | 0.160 | **46.7%/40.5%** |
| **Convex QPs (ALL)** | | | | | | | | | | |
| OSQP | -16.670 | 0.000 | 0.000 | 0.000 | 0.000 | 0.002 | - | - | - | - |
| IPOPT | -16.670 | 0.000 | 0.000 | 0.000 | 0.000 | 0.279 | 12.4 | 0.000 | 0.279 | - |
| IPM-LSTM | -16.116 | 0.000 | 0.000 | 0.003 | 0.001 | 0.044 | 8.5 | 0.157 | 0.201 | **31.2%/28.0%** |

Also, IPM-LSTM achieved the most significant reduction in iterations when the returned primal-dual solution pair is utilized for warm-starting IPOPT.

## D.2   Non-convex QPs

For non-convex QPs (8), all the non-zero elements without any special physical meaning, such as all zeros or all ones, are multiplied by a value generated from a uniform distribution in the range $[0.8, 1.2]$. If the original elements are integers, we will perform an additional rounding operation. We use "p" and "r" to represent these operations, respectively, and use "c" to denote the case without perturbation. The detailed perturbation rules for each instance are shown as:

Table 6: Perturbation rules.

| Instance | $Q_0$ | $p_0$ | $p_{\text{ineq}}$ | $q_{\text{ineq}}$ | $p_{\text{eq}}$ | $q_{\text{eq}}$ | $x^L$ | $x^U$ |
|---|---|---|---|---|---|---|---|---|
| qp1 | p | c | p | p | c | c | c | - |
| qp2 | p | c | p | p | c | c | c | - |
| st_rv1 | p | p | r | r | - | - | c | - |
| st_rv2 | p | p | r | r | - | - | c | - |
| st_rv3 | p | p | r | r | - | - | c | - |
| st_rv7 | p | p | r | r | - | - | c | - |
| st_rv9 | p | p | r | r | - | - | c | - |
| qp30_15_1_1 | p | p | p | p | p | p | c | c |

## D.3   Convex QCQPs

The performance on convex QPs, including "Convex QCQPs (RHS)" and "Convex QCQPs (ALL)", each instance with 100 variables, 50 inequality constraints, and 50 equality constraints, is shown in Table 7. From Table 7, IPM-LSTM provided the best objective value with acceptable constraint violations. Also, IPM-LSTM achieved the most significant reduction in iterations when the returned primal-dual solution pair is utilized for warm-starting IPOPT.

Table 7: Computational results on convex QCQPs

| Method | End-to-End | | | | | | IPOPT (warm start) | | Total Time (s)↓ | Gain (Ite./ Time)↑ |
|---|---|---|---|---|---|---|---|---|---|---|
| | Obj. ↓ | Max ineq. ↓ | Mean ineq. ↓ | Max eq. ↓ | Mean eq. ↓ | Time (s) ↓ | Ite. ↓ | Time (s) ↓ | | |
| **convex QCQPs (RHS)** | | | | | | | | | | |
| IPOPT | -18.761 | 0.000 | 0.000 | 0.000 | 0.000 | 0.287 | 12.3 | - | - | - |
| NN | -1.931 | 0.000 | 0.000 | 0.439 | 0.141 | <0.001 | 12.2 | 0.285 | 0.285 | 0.8%/0.7% |
| DC3 | -14.111 | 0.000 | 0.000 | 0.000 | 0.000 | <0.001 | 10.6 | 0.244 | 0.244 | 13.8%/15.8% |
| DeepLDE | -10.331 | 0.000 | 0.000 | 0.000 | 0.000 | <0.001 | 11.2 | 0.282 | 0.282 | 8.9%/1.7% |
| PDL | -15.311 | 0.006 | 0.000 | 0.005 | 0.002 | <0.001 | 10.1 | 0.227 | 0.227 | 17.9%/20.9% |
| H-Proj | -15.450 | 0.000 | 0.000 | 0.000 | 0.000 | <0.001 | 10.8 | 0.247 | 0.247 | 12.2%/13.9% |
| IPM-LSTM | -18.654 | 0.000 | 0.000 | 0.000 | 0.000 | 0.051 | 7.6 | 0.163 | 0.213 | **38.2%/25.8%** |
| **convex QCQPs (ALL)** | | | | | | | | | | |
| IPOPT | -21.849 | 0.000 | 0.000 | 0.000 | 0.000 | 0.253 | 11.8 | - | - | - |
| IPM-LSTM | -21.200 | 0.000 | 0.000 | 0.001 | 0.001 | 0.049 | 7.4 | 0.124 | 0.173 | **37.3%/31.6%** |

## D.4 A Simple Non-convex Program

$$
\begin{aligned}
\min_{x \in \mathbb{R}^n} \quad & \frac{1}{2}x^\top Q_0 x + p_0^\top \sin(x) \\
\text{s.t.} \quad & p_j^\top x \le q_j && j = 1, \cdots, l \\
& p_j^\top x = q_j && j = l+1, \cdots, m \\
& x_i^L \le x_i \le x_i^U && i = 1, \cdots, n
\end{aligned}
\tag{30}
$$

The performance on simple non-convex programs (30), including "Non-convex Programs (RHS)" and "Non-convex Programs (ALL)", each instance with 100/200 variables, 50/100 inequality constraints, and 50/100 equality constraints, is shown in Table 8. From this table, IPM-LSTM provided the best objective value with acceptable constraint violations. Also, IPM-LSTM achieved the most significant reduction in iterations when the returned primal-dual solution pair is utilized for warm-starting IPOPT.

Table 8: Computational results on non-convex programs

| Method | End-to-End | | | | | | IPOPT (warm start) | | Total Time (s)↓ | Gain (Ite./ Time)↑ |
|---|---|---|---|---|---|---|---|---|---|---|
| | Obj. ↓ | Max ineq. ↓ | Mean ineq. ↓ | Max eq. ↓ | Mean eq. ↓ | Time (s) ↓ | Ite. ↓ | Time (s) ↓ | | |
| **Non-convex Programs (RHS)**: $n = 200, m_{\text{ineq}} = 100, m_{\text{eq}} = 100$ | | | | | | | | | | |
| IPOPT | -22.375 | 0.000 | 0.000 | 0.000 | 0.000 | 0.717 | 13.1 | - | - | - |
| DC3 | -20.671 | 0.000 | 0.000 | 0.000 | 0.000 | <0.001 | 10.9 | 0.603 | 0.603 | 16.8%/15.9% |
| NN | -20.736 | 0.000 | 0.000 | 0.632 | 0.235 | <0.001 | 11.0 | 0.607 | 0.607 | 16.0%/20.7% |
| DeepLDE | -20.074 | 0.000 | 0.000 | 0.000 | 0.000 | <0.001 | 10.5 | 0.576 | 0.576 | 19.8%/19.7% |
| PDL | -21.859 | 0.589 | 0.167 | 0.026 | 0.000 | <0.001 | 10.9 | 0.600 | 0.600 | 16.8%/16.3% |
| LOOP-LC | -21.932 | 0.000 | 0.000 | 0.000 | 0.000 | <0.001 | 10.2 | 0.558 | 0.558 | 22.1%/**22.2%** |
| H-Proj | -19.097 | 0.000 | 0.000 | 0.006 | 0.000 | <0.001 | 11.5 | 0.634 | 0.634 | 12.2%/11.6% |
| IPM-LSTM | -22.213 | 0.000 | 0.000 | 0.002 | 0.001 | 0.175 | 9.5 | 0.533 | 0.708 | **27.5%**/1.3% |
| **Non-convex Programs (ALL)**: $n = 200, m_{\text{ineq}} = 100, m_{\text{eq}} = 100$ | | | | | | | | | | |
| IPOPT | -25.1043 | 0.000 | 0.000 | 0.000 | 0.000 | 0.768 | 14.3 | - | - | - |
| IPM-LSTM | -20.288 | 0.000 | 0.000 | 0.006 | 0.002 | 0.195 | 12.1 | 0.639 | 0.834 | **15.4%**/-8.6% |
| **Non-convex Programs (RHS)**: $n = 100, m_{\text{ineq}} = 50, m_{\text{eq}} = 50$ | | | | | | | | | | |
| IPOPT | -11.590 | 0.000 | 0.000 | 0.000 | 0.000 | 0.289 | 12.9 | - | - | - |
| DC3 | -10.660 | 0.000 | 0.000 | 0.000 | 0.000 | <0.001 | 11.6 | 0.259 | 0.259 | 11.6%/10.4% |
| NN | -10.020 | 0.000 | 0.000 | 0.350 | 0.130 | <0.001 | 11.4 | 0.253 | 0.253 | 11.6%/12.5% |
| DeepLDE | 4.870 | 0.000 | 0.000 | 0.008 | 0.000 | <0.001 | 13.1 | 0.294 | 0.294 | -1.6%/-1.7% |
| PDL | -11.385 | 0.006 | 0.002 | 0.001 | 0.000 | <0.001 | 9.6 | 0.207 | 0.207 | 25.6%/**28.4%** |
| LOOP-LC | -11.296 | 0.000 | 0.000 | 0.000 | 0.000 | <0.001 | 10.1 | 0.217 | 0.217 | 21.7%/24.9% |
| H-Proj | -9.616 | 0.000 | 0.000 | 0.000 | 0.000 | <0.001 | 11.3 | 0.252 | 0.252 | 12.4%/12.8% |
| IPM-LSTM | -11.421 | 0.000 | 0.000 | 0.002 | 0.001 | 0.044 | 8.9 | 0.181 | 0.225 | **31.0%**/22.1% |
| **Non-convex Programs (ALL)**: $n = 100, m_{\text{ineq}} = 50, m_{\text{eq}} = 50$ | | | | | | | | | | |
| IPOPT | -12.508 | 0.000 | 0.000 | 0.000 | 0.000 | 0.305 | 13.2 | - | - | - |
| IPM-LSTM | -12.360 | 0.000 | 0.000 | 0.001 | 0.000 | 0.044 | 8.0 | 0.149 | 0.193 | **39.4%/36.7%** |

## D.5 Performance Analysis

### D.5.1 The Number of LSTM Time Steps

The number of LSTM time steps $T$ is a key hyperparameter which decides the performance-efficiency trade-off of IPM-LSTM. To illustrate this, we conduct experiments on a convex QP (RHS) problem

with 100 variables, 50 inequality constraints, and 50 equality constraints, investigating the quality of approximate solutions under different LSTM time step settings. As shown in Table 9, the `IPM-LSTM` with deeper LSTM architectures generally yields better approximate solutions (with lower objective values, smaller constraint violation and better warm-start performance) but with longer computational time. Specifically, the `IPM-LSTM` with 60 and 70 time steps achieves most significant reductions in warm-starting iteration count and total runtime, respectively. However, taking into account the end-to-end solution time, we have opted to set $T$ to 50 in our experiments. At each IPM iteration, as the LSTM network depth increases, $\|J^k y^k + F^k\|$ decreases (see Figure 5(a)). This indicates an improvement in the quality of solutions to the linear systems. Furthermore, the corresponding `IPM-LSTM` converges faster (e.g., fewer IPM iterations) when the LSTM network becomes deeper (see Figure 4).

Table 9: Computational results on convex QPs (RHS) under different LSTM time steps.

| $T$ | End-to-End | | | | | | IPOPT (warm start) | | Total Time (s)$^\downarrow$ | Gain (Ite./ Time)$^\uparrow$ |
|---|---|---|---|---|---|---|---|---|---|---|
| | Obj. $\downarrow$ | Max ineq. $\downarrow$ | Mean ineq. $\downarrow$ | Max eq. $\downarrow$ | Mean eq. $\downarrow$ | Time (s) $\downarrow$ | Ite. $\downarrow$ | Time (s) $\downarrow$ | | |
| 10 | -12.740 | 0.000 | 0.000 | 0.006 | 0.002 | 0.014 | 9.7 | 0.203 | 0.217 | 20.5%/19.3% |
| 20 | -14.615 | 0.000 | 0.000 | 0.003 | 0.001 | 0.021 | 8.5 | 0.171 | 0.192 | 30.3%/28.6% |
| 30 | -14.753 | 0.000 | 0.000 | 0.002 | 0.001 | 0.029 | 8.0 | 0.155 | 0.184 | 34.4%/31.6% |
| 40 | -14.897 | 0.000 | 0.000 | 0.003 | 0.001 | 0.037 | 7.1 | 0.130 | 0.167 | 41.8%/37.9% |
| 50 | -14.985 | 0.000 | 0.000 | 0.001 | 0.000 | 0.045 | 6.5 | 0.115 | 0.160 | 46.7%/40.5% |
| 60 | -15.021 | 0.000 | 0.000 | 0.001 | 0.000 | 0.055 | 6.1 | 0.099 | 0.154 | 50.0%/**42.7%** |
| 70 | -15.026 | 0.000 | 0.000 | 0.000 | 0.000 | 0.064 | 6.0 | 0.100 | 0.164 | **50.8%**/39.0% |
| 80 | -15.012 | 0.000 | 0.000 | 0.000 | 0.000 | 0.073 | 6.2 | 0.106 | 0.179 | 49.2%/33.4% |
| 90 | -14.960 | 0.000 | 0.000 | 0.000 | 0.000 | 0.081 | 6.6 | 0.117 | 0.198 | 45.9%/26.4% |

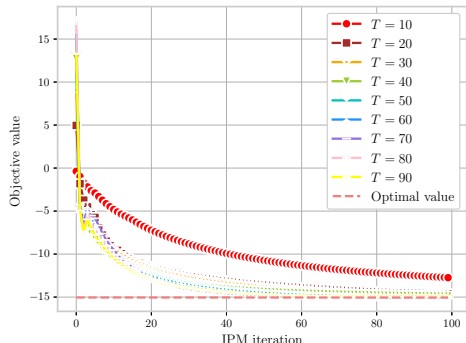

Figure 4: The objective values returned by `IPM-LSTM` at each IPM iteration on a convex QP (RHS).

### D.5.2 Linear System Solutions

In order to provide an more specific presentation of how accurately the LSTM performs, we report the detailed values of Figure 3(a) in Table 10. We can conclude that, $\|J^k y^k + F^k\|$ is roughly in the same order of magnitude as $\eta[(z^k)^\top x^k]/n$ at each IPM iteration. To reveal the relationship between the error of the linear system solution $\|J^k y^k + F^k\|$ and the LSTM time steps, hidden dimensions, training sizes and test sizes, we conduct experiments on representative convex QP (RHS) problems with 100 variables, 50 inequality constraints, and 50 equality constraints, and the results are included in Figure 5.

- In Figure 5(a), with the number of LSTM time steps increasing, $\|J^k y^k + F^k\|$ decreases.
- In Figure 5(b), we consider LSTMs with 25, 50, 75, and 100 hidden dimensions and find that an LSTM with a hidden dimension of 50, as used in our manuscript, generally performs the best (e.g., the smallest $\|J^k y^k + F^k\|$).
- In Figure 5(c), a larger training set is more beneficial for model training. The training set size in our experiment is 8,334, and the error in solving the linear system $\|J^k y^k + F^k\|$ is smaller compared with the case of 4,000 or 6,000 training samples.

- As shown by Figure 5(d), the number of samples in the test set does not affect the performance of LSTM for solving linear systems.

Table 10: The detailed values of Figure 3(a).

| IPM Ite. | $\|J^k y^k + F^k\|$ | $\eta[(z^k)^\top x^k]/n$ |
|---|---|---|
| 1 | 2.396 | 0.900 |
| 10 | 0.154 | 0.255 |
| 20 | 0.104 | 0.124 |
| 30 | 0.073 | 0.073 |
| 40 | 0.052 | 0.047 |
| 50 | 0.040 | 0.031 |
| 60 | 0.032 | 0.021 |
| 70 | 0.027 | 0.013 |
| 80 | 0.024 | 0.008 |
| 90 | 0.022 | 0.006 |
| 100 | 0.020 | 0.005 |

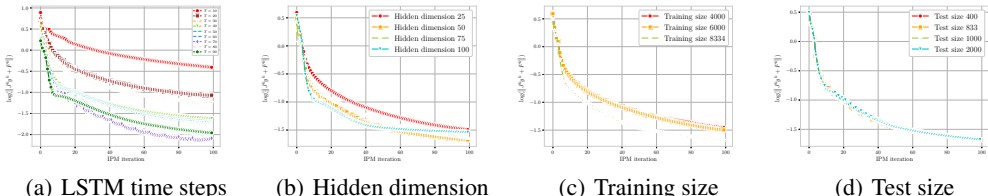

   (a) LSTM time steps     (b) Hidden dimension     (c) Training size     (d) Test size

Figure 5: The relationship between the error of the linear system solution with different parameter settings of LSTM.

We take the log of the y-axis in Figure 3(a) and plot it in Figure 6(a). Roughly speaking, $\|J^k y^k + F^k\|$ is smaller than $\eta[(z^k)^\top x^k]/n$ in the first 40 IPM iterations, while $\|J^k y^k + F^k\|$ surpasses $\eta[(z^k)^\top x^k]/n$ in the later IPM iterations. We increase the number of LSTM time steps and report the computational results in Figure 6(b). From Figure 6, we can claim that with the number of LSTM time steps increasing, $\|J^k y^k + F^k\|$ becomes smaller.

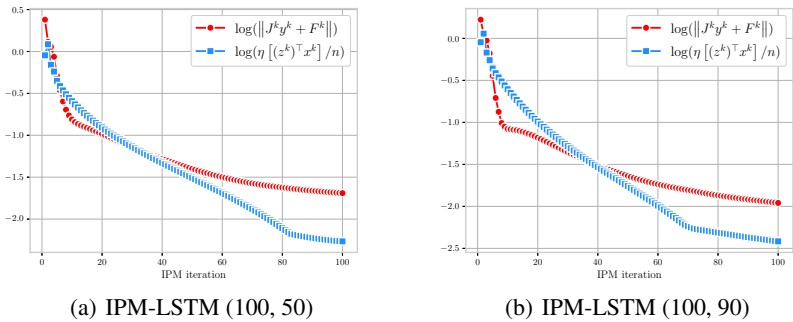

   (a) IPM-LSTM (100, 50)         (b) IPM-LSTM (100, 90)

Figure 6: Equation (5) under different LSTM time steps. The first number in the parentheses denotes the number of IPM iterations, while the second one represents the number of LSTM time steps.

### D.5.3 Condition Numbers

The LSTM approach for solving linear systems is negatively affected by their large condition numbers. To demonstrate this, we consider the least squares problem

$$\min_{y \in \mathbb{R}^l} \phi(y) := \frac{1}{2} \left\| J^k y + F^k \right\|^2 \tag{31}$$

We utilize a first-order method (e.g., steepest descent method) to minimize $\phi(y)$ and achieve a linear convergence rate (Nocedal and Wright, 1999), i.e.,

$$\phi\left(y^{t+1}\right) - \phi\left(y^\star\right) \leq \left(1 - \frac{2}{\left(\kappa\left(J^k\right)\right)^2 + 1}\right)^2 \left(\phi\left(y^t\right) - \phi\left(y^\star\right)\right) \tag{32}$$

As we discussed in Section 3.3, since solving linear systems via LSTM networks emulates iterative first-order methods, thus the value of $\kappa\left(J^k\right)$ affects the performance of LSTM networks. However, LSTM networks can empirically achieve a faster convergence rate than traditional first-order algorithms when solving the same least squares problems as shown in the computational studies (Section 3.1) of Andrychowicz et al. (2016). In order to alleviate the effect of large condition numbers, as discussed in Section 3.3, we have employed preconditioning techniques. To illustrate its effect, we conduct experiments on the simple non-convex programs, and report $\kappa\left(J^k\right)$ and their values after preconditioning (in parantheses) across several IPM iterations (e.g., $1^{st}$, $10^{th}$, $20^{th}$, $50^{th}$, $100^{th}$) in Table 11. We can conclude that the condition numbers $\kappa\left(J^k\right)$ remain within reasonable magnitudes even during the later IPM iterations, and are significantly reduced after applying the preconditioning technique.

Table 11: The condition numbers of simple non-convex programs in IPM iteration process.

| Instance | $1^{st}$ | $10^{th}$ | $20^{th}$ | $50^{th}$ | $100^{th}$ |
|---|---|---|---|---|---|
| Non-convex Programs (RHS) (100, 50, 50) | 53.8 (59.8) | 126.2 (580.7) | 153.1 (711.2) | 208.7 (1004.8) | 348.4 (1860.1) |
| Non-convex Programs (ALL) (100, 50, 50) | 55.4 (59.8) | 113.6 (517.0) | 139.8 (658.5) | 214.0 (1190.9) | 329.2 (1859.9) |
| Non-convex Programs (RHS) (200, 100, 100) | 91.5 (99.8) | 157.1 (1114.0) | 205.8 (1441.1) | 326.2 (2398.3) | 488.3 (3667.8) |
| Non-convex Programs (ALL) (200, 100, 100) | 72.1(75.7) | 175.4 (1143.4) | 184.5 (1352.7) | 249.5 (2016.6) | 368.4 (3015.3) |

