# OpenReview forum: "IPM-LSTM: A Learning-Based Interior Point Method for Solving Nonlinear Programs"
_NeurIPS.cc/2024/Conference — NeurIPS 2024 poster_

### Official Review · Reviewer_dSF8 · 2024-07-09

**Soundness:** 2
**Presentation:** 2
**Contribution:** 2
**Rating:** 5
**Confidence:** 4

**Summary:**

This paper introduces IPM-LSTM, an approach integrating Long Short-Term Memory (LSTM) neural networks with Interior Point Methods (IPMs) to solve Nonlinear Programs (NLPs). The key innovation lies in approximating solutions to linear systems within IPMs using LSTMs, aiming to accelerate the convergence of classic IPM solvers. This approach leverages the Learning to Optimize (L2O) paradigm, presenting a two-stage framework where LSTM-generated solutions are used to warm-start an IPM solver. The authors compare the IPM-LSTM against traditional solvers and recent L2O methods across various NLPs, including Quadratic Programs and Quadratically Constrained Quadratic Programs. The proposed method reportedly reduces iterations by up to 60% and solution time by up to 70%.

**Strengths:**

1. The integration of LSTM networks to approximate the solutions of linear systems in IPMs is novel.
2. The paper provides theoretical insights into the convergence properties of the proposed method under specific assumptions, adding to its credibility and understanding.
3. The paper provides a comprehensive empirical evaluation across several types of NLPs. The results demonstrate improvements over traditional methods in terms of iteration count and computational time, which supports the effectiveness of the proposed method.

**Weaknesses:**

1. The IPM-LSTM, which uses LSTM to iteratively solve linear systems at each IPM iteration, repeats K times and feeds it into the IPM solver as a warm-start point, needs more justification.
   1. The decision to use the L2O approach for solving a least squares problem (problem (4)) is not adequately justified. According to Assumption 1, the approximated solution needs to be bounded and accurate enough to guarantee the convergence of the outer loop. From the theoretical side, it is unclear how hard it is to satisfy those conditions by L2O approaches. From the empirical side, as shown in Figure 3(a), the accuracy condition is not always satisfied. If assumption 1 can not be guaranteed, the convergence of approximated IPM can not be guaranteed according to Prop. 1.
   2. The approach of using an approximated IPM solution instead of directly generating a warm-start point using either NN or L2O raises questions about efficiency and effectiveness. Previous works [1-4] have shown that direct prediction of warm-start points can be more straightforward and computationally efficient. Besides, the convergence of approximated IPM can not be guaranteed, which also raises concerns about the quality of such a warm-start point. The authors need to provide more discussion or experimental comparisons to justify their more complex, iterative approximation method.

[1] R. Sambharya, G. Hall, B. Amos, and B. Stellato, "End-to-End Learning to Warm-Start for Real-Time Quadratic Optimization", arXiv preprint arXiv:2212.08260, 2022.

[2] Sambharya R, Hall G, Amos B, et al. Learning to warm-start fixed-point optimization algorithms[J]. Journal of Machine Learning Research, 2024, 25(166): 1-46.

[3] F. Diehl, "Warm-Starting AC Optimal Power Flow with Graph Neural Networks", in Proceedings of the 33rd Conference on Neural Information Processing Systems (NeurIPS) Workshop, Vancouver, BC, Canada, Dec. 8 - 14, 2019.

[4] K. Baker, "Learning Warm-Start Points for AC Optimal Power Flow", in Proceedings of IEEE 29th Machine Learning for Signal Processing Conference, Pittsburgh, PA, USA, Oct. 13 - 16, 2019.

**Questions:**

1. In the experiments, why not directly use the simple NN prediction as a warm-start point for the IPM solver?
2. In the simulation part, as the solutions obtained by the proposed IPM-LSTM approach are used as a warm-start point to IPM, why are the equality constraints not satisfied? The authors are suggested to explain the experimental setup clearly.
3. In Table I, the proposed IPM-LSTM approach has a larger constraint violation with a longer solution time as compared to the OSQP algorithm. Does this mean the proposed approach can not exceed state-of-the-art methods for solving convex QP problems?

**Limitations:**

The authors have discussed the limitation in Sec. 5.

---

> ### Author Rebuttal · Authors · 2024-08-06
>
> Thank the reviewer for reading our manuscript  and providing constructive coments. Please refer to the Author Rebuttal for a clarification of the two-stage framework proposed in our work.
>
> ### Weakness
> >*1. The decision to use the L2O approach for solving a least squares.......*
>
> Thanks for raising this point.
> - **Theoretical**: the least-squares problem (4) in our manuscript is an **unconstrained convex optimization** problem. Several works have proved that **L2O approaches could solve such problems to guaranteed tolerances**. For example, the work of [5] proved that solutions provided by properly parameterized LSTM networks converge to one of the minimizers of composite optimization problems. Another interesting case is the work of [6] in which the authors proposed an unrolling of the ISTA algorithm and proved that solutions returned by the ISTA-unrolled method converge to the optimal ones.
> - **Empirical**: The accuracy condition is not strictly enforced in our implementation since we aim to **balance the accuracy and efficiency** of solving linear systems via LSTM. As shown in Section 4, the IPM-LSTM algorithm is able to provide high-quality approximate solutions with mild infeasibility and sub-optimality (**Stage I**). **These solutions are suitable for many practical applications**. If solutions with increased feasibility and optimality are needed, one can quickly **polish and refine** the approximate solutions by feeding them to an IPM solver (**Stage II**).
>
>
>
> >*2. The approach of using an approximated IPM solution instead of.....[1][2][3][4]...*
>
>
> Thanks for raising this point. We further clarify the motivation of our algorithmic design as follows.
>
> - In this work, we focus on developing learning-based methods to **warm-start an IPM solver for optimizing general nonlinear and convex/non-convex programs**. As the Reviewer 9Qph commented, "*The topic is important as IPM plays a crucial role in solving linear and nonlinear programs, which have extensive applications in scientific computing.*" The works of **[1][2] are focused on warm-starting fixed-point algorithms for efficiently solving convex QPs and conic programs**.
> - As we mentioned in Section 2, warm-starting an IPM solver is notoriously hard since it entails providing **well-centered primal-dual solutions** [7][8]. Recently, learning-based approaches have been proposed to address this. However, most of works such as **[3][4] did not even include dual solutions** for warm-starting. **These methods might work well for specific applications** but could not boost the performance of an IPM solver when optimizing general nonlinear programs. Some works such as [9] indeed provide primal-dual solutions but those pairs are **not intended to be well-centered**.
> - Our proposed IPM-LSTM approach works in a similar way as a classic IPM except that linear systems are approximately solved by LSTM networks. Since every iterate (e.g., primal-dual solution) follows a central path (e.g., the same parameter $\mu$ is chosen for each dimension in the perturbed KKT system, as shown in Algorithm 1), the final solution pair returned by IPM-LSTM would then be well-centered and of high quality. **Such a primal-dual solution is thus well-suited for warm-starting an IPM solver**.
>
>
> ### Questions
> >*1. ...why not directly use the simple NN prediction as a warm-start point...*
>
> Thanks for raising this point. As pointed out by [10], a **simple NN prediction often leads to significant constraint violation**. To alleviate this issue, we integrated the objective function and penalty for constraint violation into the loss function and included such a method as a baseline (denoted as "NN") in our manuscript. From Table 1, 2, 5, 7, and 8, we can conclude that **warm-starting IPOPT with initial solutions predicted by NNs brings limited (sometimes even negative) performance improvement**.
>
>
> >*2. ...why are the equality constraints not satisfied...*
>
>
> We apologize for any confusion. In the computational results, **the "IPM-LSTM" in the end-to-end fashion produces approximate solutions** that might be neither feasible nor optimal, which can result in small violations of the equality constraints. However, when **these solutions are used to warm-start an IPM solver** such as IPOPT (denoted as "IPOPT (warm-start)" in each table), IPOPT returns locally optimal solutions that **do not exhibit any constraint violations**, as shown in Table 1, 2, 5, 7, and 8.
>
>
> >*3....Does this mean the proposed approach can not exceed state-of-the-art...*
>
> Thank you for raising this question. Indeed, the proposed IPM-LSTM approach is outperformed by OSQP when solving convex QPs.
>
> - **The IPM-LSTM method is designed to provide high-quality approximate solutions for general nonlinear and convex/non-convex programs**. It follows a classic IPM approach, with the distinction that linear systems are approximately solved using an LSTM network.
> - **OSQP** employs an operator-splitting algorithm **specifically tailored for convex QPs**. This solver has demonstrated superior performance even when compared to state-of-the-art general-purpose solvers like IPOPT and Gurobi in optimizing convex QPs.
>
> Therefore, **IPM-LSTM is capable of addressing a wide range of optimization problems, but it is not as competitive as OSQP when it comes to solving convex QPs**.
>
> [5] Liu, Jialin, et al. (2023) "Towards constituting mathematical structures for learning to optimize." ICML.
>
> [6] Aberdam, A., et al. (2021). Ada-lista: Learned solvers adaptive to varying models. IEEE TPAMI.
>
> [7] Nocedal, J., & Wright, S. J. (Eds.). (1999). Numerical optimization.
>
> [8] Wright, S. J. (1997). Primal-dual interior-point methods.
>
> [9] Park, S., & Van Hentenryck, P. (2023). Self-supervised primal-dual learning for constrained optimization. AAAI.
>
> [10] Donti, P. L., et al. (2021). DC3: A learning method for optimization with hard constraints. ICLR.

---

> > ### Comment · Reviewer_dSF8 · 2024-08-10
> > **Thanks for your rebuttal**
> >
> > I have reviewed the rebuttal and acknowledge that accelerating linear system solving by NN within IPM iterations is a reasonable approach.
> >
> > While the use of NNs for solving linear systems is not new [1], integrating them within IPM solvers introduces specific challenges related to convergence and accuracy.
> >
> > I still have some concerns about the LSTM approach to solving linear systems with high accuracy.
> >
> > - The provided theoretical guarantees for the Learning to Optimize (L2O) approach are limited to convex problems.
> > - Many practical applications, especially those involving KKT systems for general non-convex problems, deal with ill-conditioned linear systems. Can the LSTM approach effectively handle such ill-conditioned systems.
> > - It is also unclear whether the used simple non-convex problem is ill-conditioned or not.
> >
> > ---
> >
> > * [1] Kaneda, A., Akar, O., Chen, J., Kala, V. A. T., Hyde, D., & Teran, J. (2023, July). A deep conjugate direction method for iteratively solving linear systems. In International Conference on Machine Learning (pp. 15720-15736). PMLR.

---

> > > ### Author Response · Authors · 2024-08-11
> > >
> > > >I have reviewed the rebuttal and acknowledge that accelerating linear system solving by NN within IPM iterations is a reasonable approach.
> > >
> > > Thanks for your efforts and acknowledgement.
> > >
> > > >While the use of NNs for solving linear systems is not new [1], integrating them within IPM solvers introduces specific challenges related to convergence and accuracy.
> > >
> > > Thanks for sharing with us this work [1]. The authors of [1] focused on using NNs to solve **positive-definite linear systems of equations** while **linear systems in IPMs are indefinite**. We incorporate this work in our revised manuscript.
> > >
> > > >I still have some concerns about the LSTM approach to solving linear systems with high accuracy.
> > >
> > >
> > > >1. The provided theoretical guarantees for the Learning to Optimize (L2O) approach are limited to convex problems.
> > >
> > > Thanks for raising this point. In this work, we focus on developing learning-based IPMs to address general nonlinear programs which might be non-convex.
> > > - The **IPM itself (i.e., the outer loop of IPM-LSTM) would guarantee that the returned solutions are locally optimal** as long as linear systems are solved to specified tolerances, as shown by Proposition 1 in our manuscript.
> > > - To solve linear systems via LSTM, we convert them to least squares problems of the form $\text{min}_y \frac{1}{2}|| J^ky+F^k ||^2$, which are **unconstrained convex programs**. As implied in Theorem 1 of [2], **solutions provided by** properly parameterized **LSTM networks would converge to one of the minimizers of convex optimization problems**.
> > >
> > > Hence, from a theoretical point of view, **the proposed IPM-LSTM approach can solve both convex and non-convex programs to optimality**. We incorporate the above points in the revised manuscript.
> > >
> > >
> > > >2. Many practical applications, especially those involving KKT systems for general non-convex problems, deal with ill-conditioned linear systems. Can the LSTM approach effectively handle such ill-conditioned systems.
> > >
> > > Thank the reviewer for raising such an interesting point. Let $\kappa(\cdot)$ denote the condition number of a matrix.
> > > - **The LSTM approach for solving linear systems is negatively affected by their large condition numbers.** To demonstrate this, we consider the least squares problem
> > > \begin{align}
> > > \underset{y \in \mathbb{R}^m}{\text{min}} \; f(y) := \frac{1}{2}|| J^ky+F^k ||^2.
> > > \end{align}
> > > We utilize a first-order method (say the steepest descent method) to minimize $f(y)$ and achieve a linear convergence rate [3], i.e.,
> > > \begin{align}
> > > f\left(x_{k+1}\right) - f\left(x^{\star}\right)   \leq   \left(1-\frac{2}{(\kappa(J^k))^2+1}\right)^{2} \left( f(x_{k})-f(x^{\star})\right).
> > > \end{align}
> > > As we discussed in the **Preconditioning** part of our manuscript, since solving linear systems via LSTM networks emulates iterative first-order methods, thus the value of $\kappa(J^k)$ affects the performance of LSTM networks.
> > > - As shown in the computational studies (Section 3.1) of [4], **LSTM networks can empirically achieve a faster convergence rate than traditional first-order algorithms** when solving the same least squares problems.
> > > - To alleviate the effect of large condition numbers, as discussed in the **Preconditioning** part of our manuscript, **we have employed preconditioning techniques**.
> > >
> > >
> > > >3. It is also unclear whether the used simple non-convex problem is ill-conditioned or not.
> > >
> > > Thanks for your suggestion. For the simple non-convex programs used in our experiment, we report $\kappa(J^k)$ and their values after preconditioning (in parantheses) across several IPM iterations (say 1, 10, 20, 50, 100) in the following table.
> > > - The condition numbers **$\kappa(J^k)$ remain within reasonable magnitudes**, even during the later IPM iterations.
> > > - Applying the **preconditioning** technique indeed significantly **reduces the condition numbers** for those non-convex problems.
> > >
> > >
> > >
> > > | Instance     | $1^{\text{st}}$ Iter. | $10^{\text{th}}$  | $20^{\text{th}}$  | $50^{\text{th}}$  | $100^{\text{th}}$ Iter. |
> > > | ------------------------- | ------- | --------- | --------- | --------- | ---------- |
> > > |**Non-convex Programs (RHS) (100, 50, 50)**|53.8(59.8)|126.2(580.7)|153.1(711.2)|208.7(1004.8)|348.4(1860.1)|
> > > |**Non-convex Programs (ALL) (100, 50, 50)**|55.4(59.8)|113.6(517.0)|139.8(658.5)|214.0(1190.9)|329.2(1859.9)|
> > > |**Non-convex Programs (RHS) (200, 100, 100)**|91.5(99.8)|157.1(1114.0)|205.8(1441.1)|326.2(2398.3)|488.3(3667.8)|
> > > |**Non-convex Programs (ALL) (200, 100, 100)**|72.1(75.7)|175.4(1143.4)|184.5(1352.7)|249.5(2016.6)|368.4(3015.3)
> > >
> > > [2] Liu, Jialin, et al. (2023) "Towards constituting mathematical structures for learning to optimize." ICML.
> > >
> > > [3] Nocedal, J., & Wright, S. J. (Eds.). (1999). Numerical optimization. New York, NY: Springer New York.
> > >
> > > [4] Andrychowicz, M., Denil, M., Gomez, S., Hoffman, M. W., Pfau, D., Schaul, T., ... & De Freitas, N. (2016). Learning to learn by gradient descent by gradient descent. Advances in neural information processing systems, 29.

---

> > > > ### Comment · Reviewer_dSF8 · 2024-08-12
> > > > **Thanks for your rebuttal**
> > > >
> > > > Thanks for your additional explanations and experiments, which address the raised concerns. I will adjust my score accordingly.
> > > >
> > > > One additional comment: for convex QCQP, why are there quadratic equality constraints? They should be linear for convex problems.

---

> > > > > ### Author Response · Authors · 2024-08-12
> > > > >
> > > > > Thanks for catching this. It is a typo in our manuscript. Indeed, all equality constraints in convex QCQPs only involve linear terms. We fix this in our revised manuscript.
> > > > >
> > > > > We really appreciate your efforts for improving our work and are happy to know that your concerns have been addressed.

---

### Official Review · Reviewer_9Qph · 2024-07-11

**Soundness:** 2
**Presentation:** 4
**Contribution:** 3
**Rating:** 7
**Confidence:** 5

**Summary:**

This paper introduces a method called IPM-LSTM, which integrates machine learning techniques into interior point methods (IPM). Specifically, the authors propose training a RNN model, LSTM, to quickly approximate the solution of linear systems within IPM. This approach is numerically validated on several convex and nonconvex QPs.

**Strengths:**

- The topic is important as IPM plays a crucial role in solving linear and nonlinear programs, which have extensive applications in scientific computing.
- The idea of accelerating a subroutine of IPM, rather than applying deep neural nets end-to-end to solve optimization problems, is insightful. The convergence of IPM is typically fast (superlinear convergence), leaving little room for improvement. However, solving the linear system in IPM is usually a computational bottleneck, making it worthwhile to accelerate with machine learning.
- The experimental results are promising. IPM-LSTM clearly outperforms other learning-based baseline methods in terms of the objective function.

**Weaknesses:**

My main concern about this paper is Assumption 1, which requires the accuracy of the linear system solution to increase as the number of iterations $k$ increases. Based on this assumption, exact convergence is derived, as shown in Proposition 1. However, the empirical results in Section 4 indicate that IPM-LSTM does not achieve exact optimality, revealing a gap between theory and practice. To address this gap, I suggest:
- Reporting the error of LSTM at each iteration. This would provide readers with an understanding of how accurately the LSTM performs. Additionally, it would be beneficial to report the relationship between the error of the linear system solution with the size, training, and testing overhead of the LSTM.
- Using a log scale y-axis for Figure 3a for better precision. For example, 0.01 is 10 times greater than 0.001, but this difference is not reflected in the linear scale of Figure 3a.
- Modifying Assumption 1 to better align with practice. For example, assuming a fixed error on the right-hand sides of equations (5) and (6). Based on this relaxed assumption, a result similar to Proposition 1 could be derived, but with a fixed error on the limit of $(x^k,\lambda^k,z^k)$. The relationship between the allowed error in Assumption 1 and the propagated eventual error in Proposition 1 would sufficiently describe the performance of IPM-LSTM.

**Questions:**

Refer to "Weaknesses".

**Limitations:**

This theoretical paper appears to have no potential negative societal impact.

---

> ### Author Rebuttal · Authors · 2024-08-06
>
> Thank the reviewer for reading our manuscript and providing constructive coments. Please refer to the Author Rebuttal for a clarification of the two-stage framework proposed in our work.
>
> > *My main concern about this paper is Assumption 1, which requires the accuracy of the linear system solution to increase as the number of iterations $k$ increases. Based on this assumption, exact convergence is derived, as shown in Proposition 1. However, the empirical results in Section 4 indicate that IPM-LSTM does not achieve exact optimality, revealing a gap between theory and practice.*
>
> Thank you for pointing this out. The IPM-LSTM utilizes an LSTM network to return approximate solutions to linear systems. Proposition 1 implies that exact optimality could be achieved by IPM-LSTM in theory. However, in practice, solving linear systems to the tolerance specified in Assumption 1 would be too time-consuming. As a result, we choose to **strike a balance between the accuracy and efficiency of solving linear systems via LSTM**. Consequently, the satisfaction of Assumption 1 is not fully ensured in our implementation, leading to the mild suboptimality and infeasibility observed in Section 4.
>
> >*To address this gap, I suggest:*
> >*1. Reporting the error of LSTM at each iteration. This would provide readers with an understanding of how accurately the LSTM performs. Additionally, it would be beneficial to report the relationship between the error of the linear system solution with the size, training, and testing overhead of the LSTM*.
>
> Thanks for your suggestion. The error of solving linear systems is indeed vital. We have plotted the progress of $\\|J^ky^k+F^k\\|$ as the IPM iteration increases in Fig. 3(a) of our manuscript. Now we report the detailed values in the following table. From this table, **$\\|J^ky^k+F^k\\|$ is roughly in the same order of magnitude as $\eta[(z^k)^{\top}x^k]/n$** at each IPM iteration.
>
> | IPM Ite. | $\|\|J^ky^k+F^k\|\|$ | $\eta[(z^k)^{\top}x^k]/n$ |
> |:------ | ----- | -----|
> | 1       |  2.396  | 0.900 |
> | 10      |  0.154  | 0.255 |
> | 20      |  0.104  | 0.124 |
> | 30      |  0.073  | 0.073 |
> | 40      |  0.052  | 0.047 |
> | 50      |  0.040  | 0.031 |
> | 60      |  0.032  | 0.021 |
> | 70      |  0.027  | 0.013 |
> | 80      |  0.024  | 0.008 |
> | 90      |  0.022  | 0.006 |
> | 100     |  0.020  | 0.005 |
>
> To reveal the relationship between the error of the linear system solution $\\|J^ky^k+F^k\\|$ and the LSTM time steps, hidden dimensions, training sizes and test sizes, we conduct experiments on representative convex QP (RHS) problems with 100 variables, 50 inequality constraints, and 50 equality constraints, and the results are included in Fig. 2 of the supplementary material.
> - In Fig. 2(a), with the LSTM time step increasing, $\\|J^ky^k+F^k\\|$ decreases.
> - In Fig. 2(b), we consider LSTMs with 25, 50, 75, and 100 hidden dimensions and find that an LSTM with a hidden dimension of 50, as used in our manuscript, generally performs the best (e.g., resulting in the smallest $\\|J^ky^k + F^k\\|$).
> - In Fig. 2\(c\), a larger training set is more beneficial for model training. The training set size used in our manuscript is 8,334, and the error in solving the linear system $\\|J^ky^k + F^k\\|$ is smaller compared with the case of 4,000 or 6,000 training samples.
> - As shown by Fig. 2(d), the number of samples in the test set does not affect the performance of LSTM for solving linear systems.
>
> >*2. Using a log scale y-axis for Figure 3a for better precision. For example, 0.01 is 10 times greater than 0.001, but this difference is not reflected in the linear scale of Figure 3a.*
>
> Thanks for your suggestion. We now take the log of the y-axis in Fig. 3(a) of our manuscript and plot it in Fig. 3(a) of the supplementary material.
> - Roughly speaking, $\\|J^k y^k + F^k\\|$ is smaller than $\eta [(z^k)^\top x^k] / n$ in the first $40$ IPM iterations, while $\\|J^k y^k + F^k\\|$ surpasses $\eta [(z^k)^\top x^k]/n$ in the later IPM iterations.
> - Following the comments from Reviewer Qdyo, we increase the LSTM time steps and report the computational results in Fig. 3(b) of the supplementary material. From Fig. 3(a) and 3(b), We can claim that **with the LSTM time steps increasing, $\\|J^k y^k + F^k\\|$ becomes smaller and closer to $\eta [(z^k)^\top x^k]/n$**.
>
> >*3. Modifying Assumption 1 to better align with practice. For example, assuming a fixed error on the right-hand sides of equations (5) and (6). Based on this relaxed assumption, a result similar to Proposition 1 could be derived, but with a fixed error on the limit of $(x^k, \lambda^k, z^k)$. The relationship between the allowed error in Assumption 1 and the propagated eventual error in Proposition 1 would sufficiently describe the performance of IPM-LSTM.*
>
> Thank you for your suggestion. **Equation (5) in our manuscript is designed to be consistent with the assumptions made in classic inexact IPMs or inexact Newton methods**, as seen in Equation (4) in [1], Equation (2.1) in [2], Equation (6) in [3], and Equation (12) in [4]. To the best of our knowledge, the theoretical bounds referenced in these works are dependent on the iteration count $k$, and do not involve fixed error terms. We would explore the use of fixed error bounds in our future studies.
>
> [1] Bellavia, Stefania. Inexact interior-point method. Journal of Optimization Theory and Applications 96 (1998): 109-121.
>
> [2] Eisenstat, Stanley C., and Homer F. Walker. Globally convergent inexact Newton methods. SIAM Journal on Optimization 4.2 (1994): 393-422.
>
> [3] Al-Jeiroudi, Ghussoun, and Jacek Gondzio. Convergence analysis of the inexact infeasible interior-point method for linear optimization. Journal of Optimization Theory and Applications 141 (2009): 231-247.
>
> [4] Gondzio, Jacek. Convergence analysis of an inexact feasible interior point method for convex quadratic programming. SIAM Journal on Optimization 23.3 (2013): 1510-1527.

---

> > ### Comment · Reviewer_9Qph · 2024-08-12
> > **Response to authors**
> >
> > I greatly appreciate the detailed response and additional experiments. My concerns are fully addressed. The new experimental results regarding the accuracy of solving linear systems look pretty promising. I would like to upgrade my score.

---

> > > ### Author Response · Authors · 2024-08-12
> > >
> > > We really appreciate your efforts for improving our work and are happy to know that all of your concerns have been addressed.

---

### Official Review · Reviewer_Qdyo · 2024-07-16

**Soundness:** 3
**Presentation:** 3
**Contribution:** 3
**Rating:** 5
**Confidence:** 4

**Summary:**

This paper proposed to replace the linear system solver used in the inner loop of interior point method (IPM) with an LSTM for solving general non-linear programs. The LSTM is trained in a unsupervised manner to minimize a unconstrained least square objective derived from the KKT conditions. The proposed framework, IPM-LSTM, can be used in an end-to-end way, or to warm-start IPM so that the number of overall outer IPM iterations can be reduces and thus the solving time decreases. The authors theoretically proved that, as long as the trained LSTM achieves certain level of accuracy, the IPM will converge. The authors conducted empirical experiments on a variety of non-linear programs to verify the effectiveness of IPM-LSTM and that it approximately satisfied the assumption in the theoretical analysis.

**Strengths:**

1. The idea of plugging in L2O methods to approximate a single step in a bigger optimization framework is interesting and promising.

2. The proposed architecture is concise and effective to some extent. The authors also provided theoretical analysis to support their method.

3. To some extent, the presented empirical resutls are promising.s

**Weaknesses:**

There are three major issues that undermine the robustness of this work:

1. The linear system solver used for IPOPT is unclear, which significantly impacts the solver's overall performance. While MUMPS is the default, HSL typically performs better, achieving 2-3 times faster results. This performance surpasses that of IPOPT when warm-started with the IPM-LSTM solution.

2. There is no analysis of how the performance changes with varying numbers of iterations in the inner LSTM, which decides the performance-efficiency trade-off of IPM-LSTM. Longer inner iterations (deeper LSTM) may improve the quality of the solutions but increases computation costs. However, improved solution quality may also decrease the number of outer iterations needed and thus reduce the overall solving time. Moreover, increased LSTM depth can make the training process more difficult. Overall, thise is tricky part and should be empirically investigated more carefully.

3. Although LSTM benefits from batched processing—a potential strength of IPM-LSTM accelerated by GPU—this work fails to provide empirical evidence supporting this property.

**Questions:**

See the weaknesses section.

**Limitations:**

See the weaknesses section.

---

> ### Author Rebuttal · Authors · 2024-08-06
>
> Thank the reviewer for reading our manuscript and providing constructive coments. Please refer to the Author Rebuttal for a clarification of the two-stage framework proposed in our work.
>
> >*1. The linear system solver used for IPOPT is unclear, which significantly impacts the solver's overall performance. While MUMPS is the default, HSL typically performs better, achieving 2-3 times faster results. This performance surpasses that of IPOPT when warm-started with the IPM-LSTM solution.*
>
> Thanks for your suggestion. The default solver MUMPS was used in our experiments. To evaluate the effect of linear solvers on the performance of IPOPT, we consider 3 commonly used linear solvers (MA27, MA57, and MA86) from HSL and bundle IPOPT with each of them. We conduct experiments on 6 datasets used in our manuscript, each with 100 variables, 50 inequality constraints, and 50 equality constraints. We report the average computational time (in seconds) for each dataset in the table below. Results for IPOPT with different linear solvers are presented in the 2nd - 5th columns, while those for IPOPT warm-started by IPM-LSTM are listed in the last four columns.
>
> - The computational results demonstrate that **MUMPS is generally outperformed by MA57** but superior to MA27 and MA86.
> - **Regardless of linear solvers, warm-starting IPOPT** with primal-dual solutions provided by **IPM-LSTM enhances the performance** of IPOPT itself.
>
>
>
> ||IPOPT||||IPM-LSTM+IPOPT||||
> |--|---|---|---|---|---|---|---|---|
> |**Dataset**|**MUMPS**|**MA27**|**MA57**|**MA86**|**MUMPS**|**MA27**|**MA57**|**MA86**|
> |**Convex QPs (RHS)**|0.269 |0.328 |0.191 |0.304 |0.170 |0.220 |**0.131** |0.195 |
> |**Non-convex Programs (RHS)**|0.289 |0.428 |0.215 |0.387 |0.225 |0.297 |**0.171** |0.299 |
> |**Convex QCQPs (RHS)**|0.287 |0.388 |0.251 |0.327 |**0.204** |0.270 |0.220 |0.226 |
> |**Convex QPs (ALL)**|0.279 |0.354 |0.199 |0.376 |0.201 |0.272 |**0.159** |0.245 |
> |**Non-convex Programs (ALL)**|0.305 |0.396 |0.213 |0.387 |0.193 |0.256 |**0.146** |0.237|
> |**Convex QCQPs (ALL)**|0.253 |0.328 |0.213 |0.311 |0.173 |0.202 |**0.160** |0.194 |
>
>
> >*2. There is no analysis of how the performance changes with varying numbers of iterations in the inner LSTM, which decides the performance-efficiency trade-off of IPM-LSTM. Longer inner iterations (deeper LSTM) may improve the quality of the solutions but increases computation costs. However, improved solution quality may also decrease the number of outer iterations needed and thus reduce the overall solving time. Moreover, increased LSTM depth can make the training process more difficult. Overall, thise is tricky part and should be empirically investigated more carefully.*
>
> Thanks for your suggestion. We first remark that in our implementation, the number of IPM iteration (e.g., outer iteration) is fixed to $100$. The number of iterations in the inner LSTM indeed incurs the performance trade-off. To illustrate this, we conduct experiments on convex QP (RHS) problems with 100 variables, 50 inequality constraints, and 50 equality constraints, investigating the quality of approximate solutions under different LSTM inner iteration settings. We report the results in the table below and Fig. 1 in the supplementary material.
> - At each IPM iteration, **as the LSTM network depth increases**, $\\|J^k y^k + F^k\\|$ decreases (see Fig. 1(a)). This indicates **an improvement in the quality of solutions to the linear systems**. Furthermore, the corresponding **IPM-LSTM converges faster** (e.g., fewer outer iterations) when the LSTM network becomes deeper (see Fig.1(b)).
> - From the table below, generally speaking, **the IPM-LSTM with deeper LSTM architectures tends to produce better approximate solutions** (with lower objective values and smaller constraint violation) but **with longer computational time.**
> - Training deeper LSTM networks indeed becomes more challenging, such as **longer training time and more memory consumption** due to large computational graphs, **vanishing or exploding gradient** issues [1].
>
>
> | # Ite. in LSTM | Obj. | Max ineq. | Mean ineq. | Max eq. | Mean eq. | Time (s) |
> | -------- | -------- | -------- | -------- | -------- | -------- | -------- |
> |10   | -12.740 | 0.000  | 0.000 |  0.006  |  0.002   | 0.012 |
> |20   | -14.615 | 0.000  | 0.000 |  0.003  |  0.001   | 0.020 |
> |30   | -14.753 | 0.000  | 0.000 |  0.002  |  0.001   | 0.028 |
> |40   | -14.897 | 0.000  | 0.000 |  0.003  |   0.001  | 0.037  |
> |50   | -14.906 | 0.000  | 0.000 | 0.001   |  0.000   | 0.045 |
> |60   | -15.021 | 0.000  | 0.000 |  0.001  |  0.000   | 0.055  |
> |70   | -15.026 | 0.000  | 0.000 |  0.000  |  0.000   | 0.063 |
> |80   | -15.012 | 0.000  | 0.000 |  0.000  |  0.000   | 0.072 |
> |90   | -14.960 | 0.000  | 0.000 |  0.000  |  0.000   | 0.080 |
>
> >*3. Although LSTM benefits from batched processing—a potential strength of IPM-LSTM accelerated by GPU—this work fails to provide empirical evidence supporting this property.*
>
> Thanks for your suggestion. Given a batch size of $k$ testing samples, we are able to feed all of them to the trained IPM-LSTM at the same time. Let $T$ denote the wall-clock time used when all samples are solved by IPM-LSTM. The average solution time for each instance is then $T/k$. Thus, the proposed IPM-LSTM approach can indeed leverage the advantage of GPU batch processing.
>
> To demonstrate this, we conducted tests on convex QPs (RHS) with 100 variables, 50 inequalities, and 50 equalities, with batch sizes varying from 10 to 5,000. The computational results are reported in Fig. 1\(c\) of the supplementary material. As shown in Fig. 1\(c\), **the average solution time decreases as the batch size increases**. However, due to hardware limitations, once the batch size exceeds a certain threshold, the average computational time will level off.
>
> [1] Pascanu, Razvan, Tomas Mikolov, and Yoshua Bengio. "On the difficulty of training recurrent neural networks." International conference on machine learning. PMLR, 2013.

---

> ### Comment · Area_Chair_HVao · 2024-08-13
> **Please Engage in Discussion**
>
> Dear Reviewer,
>
> Thank you for your time and efforts throughout the review period. Please read the authors' rebuttal as soon as possible and indicate if their responses have addressed all your concerns.
>
> Best,
>
> Your AC

---

### Author Rebuttal · Authors · 2024-08-06

Before addressing the reviewers' comments, we would like to first clarify the two-stage framework proposed in our work.
- **Stage I**: we **utilize IPM-LSTM to produce a high-quality approximate solution** (which might neither be feasible nor optimal but presumably well-centered). The IPM-LSTM is designed like a classic IPM with linear systems being solved by an LSTM network rather than a linear system solver.
- **Stage II**: we then use the approximate solution to **warm start an IPM solver**, such as IPOPT.

To the best of our knowledge, **our work is the first attempt of enhancing IPMs with learning-based techniques for addressing general nonlinear programs**.

---

### Decision · Program_Chairs · 2024-09-25

**Decision:**

Accept (poster)

**Comment:**

This paper proposed to replace the linear system solver used in the inner loop of the interior point method (IPM) with an LSTM for solving general non-linear programs. Most of the reviewers agreed that this paper is technically solid. Overall, this paper is well-structured with a logical flow, and the idea of integrating machine learning techniques into interior point methods is valuable for this community. All the reviews tend to accept the paper during the reviewer-author discussion phase. Therefore, I recommend this paper to the NeurIPS 2024 conference.